# Cholecystokinin modulates age-dependent thalamocortical neuroplasticity

Xiao Li[1,2,3,4]*†, Jingyu Feng[2,5]†, Xiaohan Hu[1,2,3]†, Peipei Zhou[1,2,3], Tao Chen[1,2,3,6], Xuejiao Zheng[1,2,3,6], Peter Jendrichovsky[2], Xue Wang[1,2,3], Mengying Chen[1,2,3], Hao Li[1,2], Xi Chen[1,2,3], Dingxuan Zeng[1,2,3], Mengfan Zhang[1,2,3,6], Zhoujian Xiao[1,2,3], Ling He[1,2,3,4,6], Stephen Temitayo Bello[1,2,3], Jufang He[1,2,3,4,6]*

[1]Department of Neuroscience, City University of Hong Kong, Hong Kong, Hong Kong; [2]Department of Biomedical Science, City University of Hong Kong, Hong Kong, Hong Kong; [3]Research Centre for Treatments of Brain Disorders, City University of Hong Kong, Hong Kong, Hong Kong; [4]CAS Key Laboratory of Brain Connectome and Manipulation, the Brain Cognition and Brain Disease Institute, Shenzhen Institute of Advanced Technology, Chinese Academy of Sciences, Beijing, China; [5]Department of Neuroscience, University of Rochester Medical Center, Rochester, United States; [6]Center of Regenerative Medicine and Health, Hong Kong Institute of Science and Innovation, Chinese Academy of Sciences, Beijing, China

*For correspondence:
xli293@cityu.edu.hk (XL);
jufanghe@cityu.edu.hk (JH)

†These authors contributed equally to this work

Competing interest: The authors declare that no competing interests exist.

## eLife Assessment

This is an **important** study demonstrating that cholecystokinin is a key modulator of auditory thalamocortical plasticity during development and in young adult but not aged mice, though cortical application of this neuropeptide in older animals appears to go some way to restoring this age-dependent loss in plasticity. A strength of this work is the use of multiple experimental approaches, which together provide **convincing** support for the proposed involvement of cholecystokinin. This work is likely to be influential in opening up a new avenue of investigation into the roles of neuropeptides in sensory plasticity.

**Abstract** The thalamocortical pathways exhibit neuroplasticity not only during the critical period but also in adulthood. In this study, we investigated how cholecystokinin (CCK) modulates age-dependent thalamocortical plasticity. Our findings demonstrated that CCK is expressed in thalamocortical neurons and that high-frequency stimulation (HFS) of the thalamocortical pathway triggers the release of CCK in auditory cortex (ACx), as detected by a CCK sensor. HFS of the medial geniculate body (MGB) induced thalamocortical long-term potentiation (LTP) in wild-type young adult mice. However, knockdown of CCK expression in MGB neurons or blockade of the CCK-B receptor (CCKBR) in the ACx abolished HFS-induced LTP. Interestingly, this LTP could not be elicited in juvenile (3-week-old) or aged mice (over 18-month-old) due to distinct mechanisms: the absence of CCKBR in juveniles and the inability to release CCK in aged mice. Notably, exogenous administration of CCK into the ACx rescued LTP in aged mice and significantly improved frequency discrimination. These findings highlight the potential of CCK as a therapeutic intervention for ameliorating neuroplasticity deficits associated with thalamocortical connectivity.

## Introduction

Long-term potentiation (LTP) in the thalamocortical pathway is crucial for the establishment and refinement of topographical maps in sensory cortices during development. In vitro studies have shown that thalamocortical LTP can be induced by pairing pre- and postsynaptic activation during the first postnatal week (*Crair and Malenka, 1995*). In the auditory cortex (ACx), neonatal exposure to tones during early postnatal days induces significant alterations in the tonotopic map (*Zhang et al., 2002*). This frequency-specific plasticity is mediated by precise thalamocortical inputs that relay frequency information to the cortex, with LTP and LTD being proposed as the synaptic mechanisms underlying this process (*Liu et al., 2011*). Towards the end of the critical period for topographical map formation in each sensory modality, neonatal thalamocortical LTP disappears due to a developmental switch in NMDA receptor subunits, which decreases the sensitivity of sensory cortices to passive input (*Barth and Malenka, 2001*; *Malenka and Bear, 2004*; *Liu et al., 2004*).

However, converging evidence shows that thalamocortical inputs retain a capacity for experience-dependent modification in adulthood (*Yu et al., 2012*; *Biane et al., 2016*; *Audette et al., 2019*; *Oberlaender et al., 2012*). Sensory enrichment or deprivation can gate or reinstate thalamocortical plasticity (*Montey and Quinlan, 2011*; *Mainardi et al., 2010*; *Rodríguez et al., 2018*; *Zhou et al., 2011*; *Blundon et al., 2017*; *Chung et al., 2017*; *Dahmen and King, 2007*). In the adult ACx, pairing sounds with neuromodulatory drive can reshape cortical representations (*Chun et al., 2013*). In vivo high-frequency stimulation (HFS) of dorsal lateral geniculate nucleus (LGN) or medial geniculate body (MGB) induces LTP in sensory cortices and has been linked to perceptual learning beyond the critical period (*Hogsden and Dringenberg, 2009*; *Speechley et al., 2007*; *Heynen and Bear, 2001*; *Cooke and Bear, 2010*). Notably, auditory thalamocortical plasticity appears less dependent on NMDA receptors compared to other cortical regions (*Chun et al., 2013*). The mechanisms underlying thalamocortical plasticity in the mature brain remain poorly understood.

Cholecystokinin (CCK) and its receptor CCK-B receptor (CCKBR) are well positioned to influence thalamocortical transmission: *Cck* mRNA is abundant in MGB neurons and CCKBR is enriched in layer IV of ACx, the principal thalamorecipient layer (*Zarbin et al., 1983*; *Senatorov et al., 1997*). Previous studies have highlighted that CCK, acting through CCKBR, plays a critical role in cortical plasticity (*Li et al., 2014*; *Chen et al., 2019a*). CCKBR activation enhances intracellular calcium level, which is crucial for LTP induction (*Li et al., 2023*). The presence of CCK facilitates synaptic potentiation following presynaptic and postsynaptic neuronal activity (*Li et al., 2014*). Moreover, HFS has been shown to induce cortical LTP in a CCK-dependent manner (*Chen et al., 2019a*; *Li et al., 2023*). HFS of the entorhino-neocortical pathway triggers CCK release in the ACx, subsequently leading to cortico-cortical LTP and the formation of sound-sound associative memory (*Chen et al., 2019a*; *Li et al., 2023*), as well as visuoauditory associative memory (*Zhang et al., 2020*; *Sun et al., 2024*). Pharmacological blockade of CCKBR inhibits HFS-induced neocortical LTP and disrupts associative memory formation (*Li et al., 2014*; *Chen et al., 2019a*).

In the present study, we hypothesize that CCK, expressed in the MGB, plays an essential role in the formation of thalamocortical LTP in the adult brain. We propose that the expression of CCK and CCKBR correlate with the emergence of thalamocortical LTP at distinct stages of postnatal development. To test this hypothesis, we investigated the existence of HFS-induced thalamocortical LTP while specifically downregulating CCK expression in MGB neurons or blocking CCKBR with an antagonist in the ACx. Using optogenetics, we selectively activated CCK-positive thalamocortical neurons to induce LTP. We also employed a specialized CCK sensor to assess CCK release triggered by HFS. Additionally, we used RNAscope in situ hybridization and immunohistochemistry to explore the relationship between thalamocortical LTP and the ontogeny of CCK and CCKBR expression at different life stages. Finally, we investigated whether exogenous CCK application could rescue thalamocortical LTP deficits in aged animals and whether the restored thalamocortical neuroplasticity could improve sound discrimination in behavioral tests.

## Results

### HFS-induced thalamocortical LTP enhances neuronal responses to a natural stimulus in the ACx

In the present study, we applied HFS to the MGB in an in vivo preparation to induce thalamocortical LTP and investigated its effects on auditory cortical responses in young adult mice (8-week-old). Stimulation electrodes were placed in the MGB (specifically in the medial geniculate nucleus ventral subdivision, MGv), and recording electrodes were inserted into layer IV of ACx (*Figure 1A*). Higher-magnification histology confirmed accurate MGv targeting (*Figure 1A*, lower-middle panel). Both electrodes effectively recorded neuronal responses to noise bursts (*Figure 1B*). Field excitatory post-synaptic potentials (fEPSPs) in response to electrical stimulation (ES) of the MGB were measured. Consistent with previous studies (*Speechley et al., 2007*; *Heynen and Bear, 2001*), HFS (*Figure 1C*) successfully induced thalamocortical LTP in young adult mice, as evidenced by a 30.0 ± 4.8% increase in fEPSP slopes 1 hr after HFS compared to baseline recorded before HFS (*Figure 1D*, one-way RM ANOVA, F[1,12]=39.4; pairwise comparison, before vs. after HFS: 100.7±0.4% vs 130.7 ± 4.8%, p<0.001, n=13 different recording and stimulation sites, from N=10 mice).

Given that thalamocortical LTP enhances synaptic responses to ES, we examined whether neuronal responses to a natural auditory stimulus in the ACx were also potentiated after HFS-induced thalamocortical LTP (*Figure 1E*). As shown in the example, the firing rate of ACx neurons in response to noise bursts increased following HFS in the MGB (*Figure 1F*). Additionally, fEPSPs evoked by noise bursts showed significant potentiation after HFS (*Figure 1G*, one-way RM ANOVA, F[1,16]=67.0; pairwise comparison, before vs. after HFS, 98.9±1.7% vs 135.4 ± 2.9%, p<0.001, n=17, from 6 mice), indicating that HFS-induced thalamocortical LTP facilitates cortical responses to natural auditory stimuli. Importantly, these experiments were conducted in adult mice, confirming that thalamocortical plasticity, induced by HFS, persists in the auditory system beyond the critical period.

### HFS-induced thalamocortical LTP is CCK-dependent

Previous studies have shown that both exogenous and endogenous CCK can induce synaptic plasticity in various neural pathways (*Li et al., 2014*; *Chen et al., 2019a*; *Li et al., 2023*; *Zhang et al., 2020*; *Sun et al., 2024*; *Feng et al., 2021*). HFS of entorhino-neocortical terminals triggers the release of CCK, facilitating the formation of cortical LTP (*Chen et al., 2019a*; *Li et al., 2023*). Moreover, CCK expression is abundant in MGB neurons (*Senatorov et al., 1997*; *Burgunder and Young, 1988*; *Fallon and Seroogy, 1984*), and high densities of CCK receptors are specifically located in layer IV of the ACx, the primary target of thalamocortical afferents (*Zarbin et al., 1983*). Based on these findings, we hypothesized that the HFS-induced thalamocortical LTP is CCK-dependent.

To directly test whether HFS of the thalamocortical CCK projections induces homosynaptic LTP, we employed optogenetics to selectively activate CCK-positive projections. Cre-dependent AAV9-EFIa-DIO-ChETA-EYFP was injected into the MGB of *Cck*^Cre mice. EYFP labeling marked CCK-positive neurons in the MGB. The co-expression of EYFP thalamocortical projections with PSD95 confirms the identity of thalamocortical terminals (yellow), which primarily targeted layer IV of the ACx (*Figure 2A*, upper panel). Immunohistochemistry revealed that a substantial proportion (15 out of 19, *Figure 2A* lower right panel) of thalamocortical terminals (arrows) colocalize with CCK receptors (CCKBR) on postsynaptic cortical neurons in the ACx (*Figure 2A* lower panel), supporting the functional role of CCK in modulating thalamocortical plasticity. During the optogenetic electrophysiology experiment, a glass recording electrode was inserted into layer IV of the ACx, and an optic fiber was positioned near the recording electrode to activate thalamocortical terminals with laser stimulation (*Figure 2B* upper left, and *Figure 2—figure supplement 1A*). Following a stable 16 min baseline recording of laser-evoked thalamocortical fEPSPs, high-frequency laser stimulation (HFLS) was delivered to the ACx. The fEPSPs evoked by HFLS exhibited reliable synchronization with thalamocortical afferent activation at 80 Hz laser pulses (*Figure 2—figure supplement 1B*). Notably, robust LTP was observed after HFLS (*Figure 2B*, one-way RM ANOVA, F[1,24]=62.8; pairwise comparison, before vs. after HFLS-ACx: 100.2±0.7% vs 131.0 ± 3.8%, p<0.001, n=25 different recording and stimulation sites, from 5 mice). Similarly, HFLS applied to MGB cell bodies induced significant potentiation of fEPSPs in the ACx (*Figure 2—figure supplement 1C*, one-way RM ANOVA, F[1,17]=31.8; pairwise comparison, before vs. after HFLS-MGB: 99.6±1.4% vs 130.0 ± 4.7%, p<0.001, n=18, from 6 mice). These results support

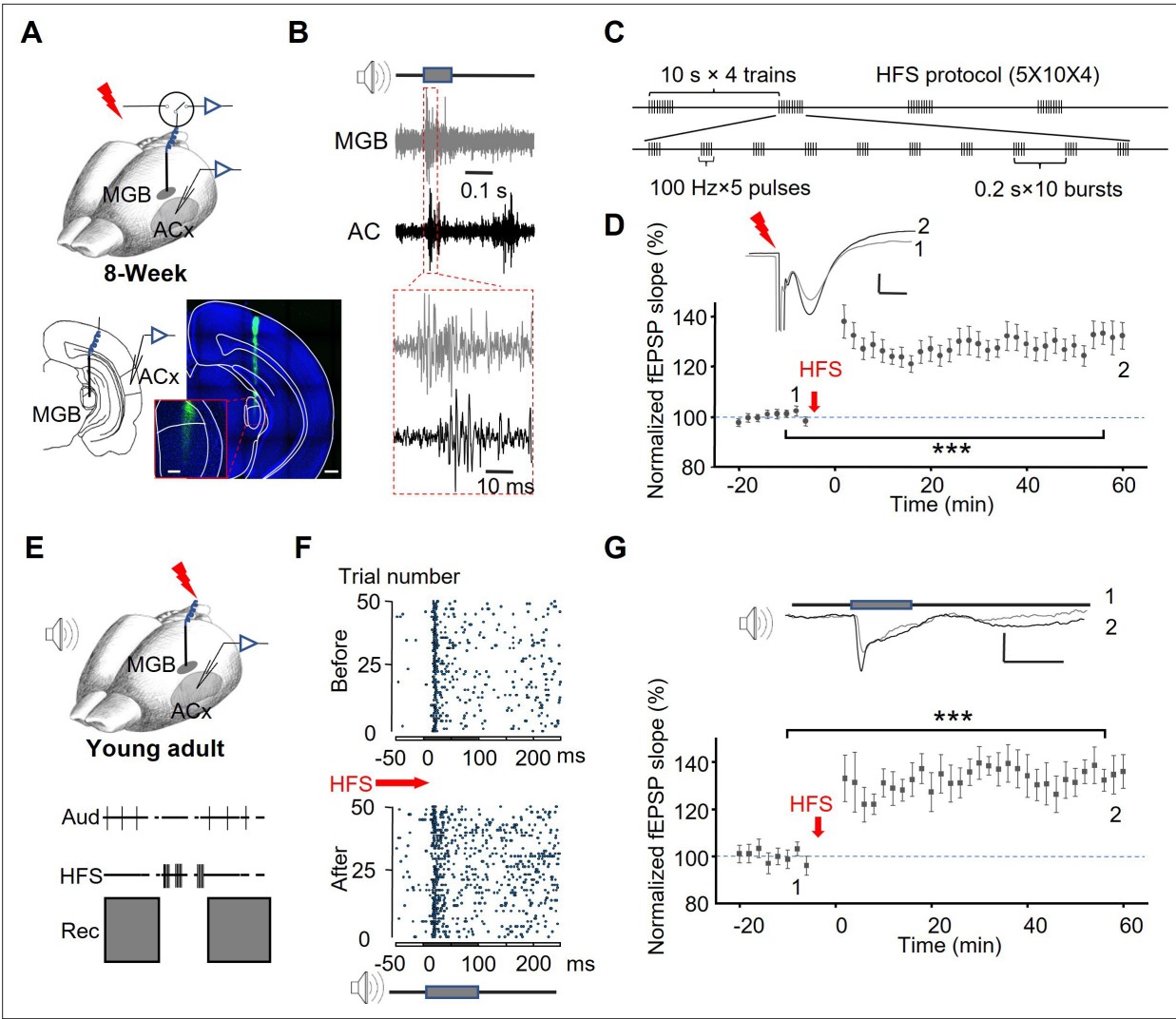

**Figure 1.** HFS of the MGB induces thalamocortical LTP in the ACx. (**A**) Schematic of the experimental setup. Stimulation electrodes were inserted in the MGB, and the recording electrodes were placed into the ACx. Post-hoc histology at higher magnification (lower-middle) shows the electrode tip confined within the MGv. White lines delineate the MGv/MGd border based on cytoarchitectonic landmarks. Scale bars: Lower-middle, 100 μm; Right, 400 μm. (**B**) Representative raw extracellular multiunit recordings from the MGB and ACx. Noise bursts of 100ms duration and 70 dB intensity were presented during electrode insertion. Robust auditory responses were recorded when the electrodes reached layer IV of the ACx and the ventral division of MGB. (**C**) HFS protocol. Each burst consisted of five 0.5 ms pulses at 100 Hz. Each block included 10 bursts at 5 Hz, with an inter-block interval of 10 s. Four blocks were delivered in total. (**D**) HFS-induced LTP in the auditory thalamocortical pathway. Upper: Representative waveforms of fEPSPs evoked by ES in MGB, with pre-HFS responses shown in gray (labeled 1) and post-HFS responses shown in black (labeled 2). Scale bar: 10ms, 0.1 mV. Lower: Population data of normalized fEPSP slopes recorded in the ACx before and 1 hr after HFS (n=13 recording sites from N=10 young adult mice, One-way RM ANOVA, ***, p<0.001). Data are mean ± SEM; error bars indicate SEM. (**E**) The preparation (upper) and the paradigm for induction of LTP in natural auditory responses (lower). (**F**) Raster plots showing auditory responses in the ACx before and after thalamocortical LTP induction. (**G**) HFS-induced LTP of auditory responses. Upper: Representative waveforms of fEPSPs evoked by noise bursts before (Gray, 1) and after (Black, 2) HFS in the MGB. Scale bar: 0.1 s, 0.2 mV. Lower: Population data of normalized fEPSP slopes recorded in the ACx before and after HFS (n=17 from 6 mice, One-way RM ANOVA, ***, p<0.001).

the notion that high-frequency activation of thalamocortical CCK projections induces thalamocortical LTP, likely through homosynaptic CCK release.

To further substantiate the dependence of HFS-induced thalamocortical LTP on CCK, we specifically knocked down CCK expression in the MGB neurons by injecting AAV constructs carrying a short hairpin RNA (shRNA) targeting *Cck* (anti-*Cck*) or a nonsense sequence (anti-Scramble; *Figure 2C*). Post-hoc validation confirmed successful downregulation of *Cck* mRNA levels by anti-*Cck* shRNAs in vivo (*Feng et al., 2021*). No discernible differences in baseline ACx responses to MGB stimulation

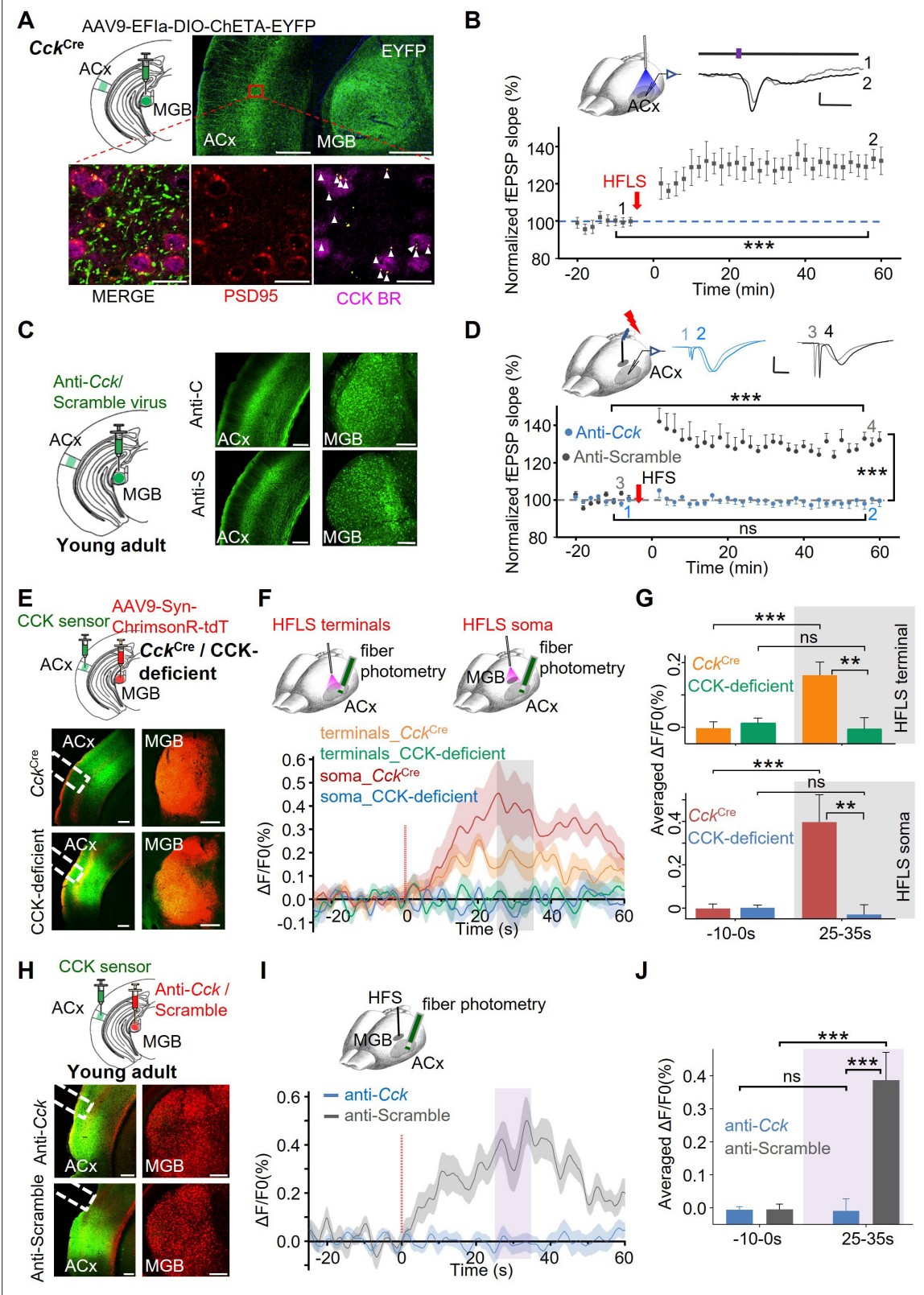

**Figure 2.** Thalamocortical LTP is CCK-dependent. (**A**) Upper panel: Cre-dependent expression of EYFP in CCK-positive thalamocortical projection neurons in *Cck*^Cre mice. Histological validation shows viral expression in the ACx and MGB (Green: EYFP). Lower panel: The post-synaptic marker PSD95 was labeled in red. CCK-B receptors are labeled by far-red-fluorescent dye and represented in magenta. Thalamocortical terminals (yellow dots) and their co-localization with CCKBR are indicated by white arrowheads. Scale bars: Upper, 300 μm; Lower, 20 μm. (**B**) Induction of LTP in the thalamocortical

*Figure 2 continued on next page*

*Figure 2 continued*

pathway by HFLS of CCK-positive thalamocortical fibers in the ACx. Upper left: Experiment design schematic. Upper right: Representative waveforms of laser-evoked fEPSPs before (Gray, 1) and after (Black, 2) HFLS. Scale bar: 40ms, 20 μV. Lower panel: Normalized slopes of laser-evoked fEPSPs for 16 min before and 1 hr after HFLS in the ACx (n=25 from 5 mice, one-way RM ANOVA, ***, p<0.001). (**C**) Left panel: Schematic diagram of viral injections. Mice were injected with AAV expressing shRNA targeting *Cck* (anti-*Cck*: rAAV-hSyn-EGFP-5′miR-30a-shRNA(*Cck*)–3′-miR30a-WPREs) or a scrambled control sequence (anti-Scramble: rAAV-hSyn-EGFP-5′miR-30a-shRNA(Scramble)–3′-miR30a-WPREs). Right panel: Post-hoc verification of viral expression in the MGB and thalamocortical projections distributed in the ACx. Scale bars: 200 μm. (**D**) Upper left: Schematic diagram of the experimental design. Upper right: Representative fEPSP traces before (1,3) and after (2,4) HFS in the anti-Scramble (gray) and anti-*Cck* (blue) groups. Scale bar: 10ms, 0.1 mV. Lower panel: Normalized fEPSP slopes before and after HFS in the anti-*Cck* group (blue, ns, p=0.859, n=16 from 10 mice) and anti-Scramble group (gray, ***, p<0.001, n=17 from 8 mice; two-way ANOVA comparison between groups after HFS:, ***, p<0.001). (**E**) Upper panel: AAV9-syn-CCKsensor was injected into the ACx, and AAV9-Syn-FLEX-ChrimsonR-tdTomato/AAV9-Syn-ChrimsonR-tdTomato was injected into the MGB of CCK-ires-Cre/CCK-deficient mice. Lower panel: Enlarged images showing the expression of AAV9-hSyn-ChrimsonR-tdTomato in the MGB, its thalamocortical projections to the ACx, along with the expression of CCK-sensor in the ACx. Scale bars: 200 μm. (**F**) Upper: Schematic diagram of the experimental design. An optical fiber was attached to the surface of ACx to activate thalamocortical terminals (left) or inserted into the MGB to activate the cell body (right). Another optical fiber was placed in the ACx to monitor fluorescence intensity. Lower: Traces of fluorescence signals of the CCK-sensor before and after HFLS (red light) in *Cck*^Cre^ or CCK-deficient mice. The averaged ΔF/F0% is presented as solid lines, with the SEM indicated by the shadow area. Fluorescence increased after HFLS in *Cck*^Cre^ mice (Orange, terminal activation; red, soma activation), whereas HFLS failed to induce an increase in CCK-deficient mice (green, terminal activation; blue, soma activation). (**G**) Bar charts showing averaged ΔF/F0% from different groups before and after HFLS (Bonferroni multiple comparisons adjustment: Upper, before vs. after HFLS at terminal in *Cck*^Cre^: ***, p<0.001, n=16, N=8; before vs. after HFLS at terminal in CCK-deficient: ns, p=0.616, n=14, N=7; After HFLS at terminal in *Cck*^Cre^ vs. CCK-deficient: **, p=0.004; Lower, before vs. after HFLS at soma in *Cck*^Cre^: ***, p<0.001, n=21, N=11; before vs. after HFLS at soma in CCK-deficient: ns, p=0.723, n=20, N=10; After HFLS at soma in *Cck*^Cre^ vs. CCK-deficient: **, p=0.003). (**H**) Upper: Schematic diagram of viral injections. AAV9-syn-CCKsensor was injected into ACx, and anti-*Cck* (rAAV-hSyn-mCherry-5′miR-30a-shRNA(*Cck*)–3′-miR30a-WPREs) /anti-Scramble (rAAV-hSyn-mCherry-5′miR-30a-shRNA(Scramble)–3′-miR30a-WPREs) shRNAs was injected into the MGB of C57 mice. Lower: Enlarged images show the expression of shRNAs in the MGB, its thalamocortical projections to the ACx, along with the expression of CCK sensor in the ACx. Scale bars: 200 μm. (**I**) Upper: Experiment design schematic. A stimulation electrode was inserted into MGB for HFS. Another optical fiber was placed in ACx to monitor the fluorescence intensity. Lower: Traces of fluorescence signals of the CCK-sensor before and after HFS in the anti-*Cck* group (blue) or anti-Scramble group (gray). The averaged ΔF/F0% is presented as solid lines, with the SEM indicated by the shadow area. Fluorescence increased after HFS of MGB in the anti-Scramble group (gray), whereas the HFS failed to induce an increase in the anti-*Cck* group (blue). (**J**) Bar charts showing the averaged ΔF/F0% from different groups before and after HFS (Bonferroni multiple comparisons adjustment: before vs. after HFS in anti-Scramble group: ***, p<0.001, n=21 from 11 mice; before vs. after HFS in anti-*Cck* group: ns, p=0.999, n=22 from 11 mice; After HFLS in anti-Scramble group vs. anti-*Cck* group: ***, p<0.001).

The online version of this article includes the following figure supplement(s) for figure 2:

**Figure supplement 1.** Thalamocortical LTP is CCK-dependent.

---

were observed between anti-*Cck* and anti-Scramble groups (***Figure 2D*** upper middle). Notably, the anti-*Cck* group exhibited no induction of thalamocortical LTP following HFS in the MGB, whereas the anti-Scramble control group showed significant LTP (***Figure 2D***, two-way RM ANOVA, F[1,31]=50.3, p<0.001; pairwise comparison: anti-*Cck*, before vs. after HFS: 99.7±0.5% vs 99.2 ± 2.9%, p=0.859, n=16 from 10 mice; anti-Scramble, before vs. after HFS: 100.8±0.5% vs 129.4 ± 2.8%, p<0.001, n=17 from 8 mice; The difference between groups after HFS: 30.2 ± 4.0%, p<0.001). We next investigated whether blocking CCKBR in the ACx would similarly impair HFS-induced thalamocortical LTP. L-365,260 (L365; 250 nM, 0.5 μL), a selective CCKBR antagonist, was administered into the ACx prior to HFS in the MGB. Artificial cerebrospinal fluid (ACSF) was used as a control. HFS-induced thalamocortical LTP was abolished in the L365 group but remained intact in the ACSF group (***Figure 2—figure supplement 1D***, two-way RM ANOVA, F[1,32]=29.3, p<0.001; pairwise comparison: L365 group, before vs. after HFS, 99.5±0.6% vs 100.4 ± 3.2%, p=0.779, n=22 from 5 mice; ACSF group, before vs. after HFS, 100.3±0.8% vs 128.9 ± 4.3%, p<0.001, n=12 from 5 mice; The difference between groups after HFS: 28.5 ± 5.4%, p<0.001). Taken together, these results provide strong evidence that HFS-induced thalamocortical LTP critically depends on presynaptic CCK and its activation of postsynaptic CCKBR receptors in the ACx.

Previous studies have demonstrated that high-frequency activation of entorhinal neurons or entorhino-neocortical CCK projections elicits CCK release from their terminals (***Chen et al., 2019a***; ***Li et al., 2023***). To determine whether HFS of the MGB or thalamocortical projections induces CCK release in the ACx, we utilized a G-protein-coupled receptor (GPCR) activation-based CCK sensor (***Wang et al., 2023***). This sensor detects CCK binding to CCKBR, resulting in increased fluorescence intensity, which can be measured via fiber photometry. We injected the AAV9-syn-CCK-sensor virus into the ACx and AAV9-Syn-FLEX-ChrimsonR-tdTomato/AAV9-Syn-ChrimsonR-tdTomato into the MGB of

$Cck^{Cre}$/CCK-deficient mice (**Figure 2E** upper panel). After 6 weeks of viral expression (**Figure 2E** lower panel), HFLS (620 nm) was applied to thalamocortical terminals in the ACx. Fluorescence intensity markedly increased in $Cck^{Cre}$ mice, while no significant changes were observed in CCK-deficient mice (**Figure 2F and G** upper panel, green and orange traces, Averaged ΔF/F0%: two-way ANOVA with Bonferroni multiple comparisons adjustment, significant interaction F[1, 28]=13.25, p=0.001; before vs. after HFLS in $Cck^{Cre}$: −0.002±0.018 vs 0.163±0.037, p<0.001, n=16 from 8 mice; before vs. after HFLS in CCK-deficient: 0.015±0.019 to -0.004±0.039, ns, p=0.616, n=14 from 7 mice; After HFLS in $Cck^{Cre}$ vs. CCK-deficient: 0.163±0.037 to -0.004±0.039, p=0.004). We next directly applied HFLS to the MGB. A substantial increase in fluorescence intensity was observed in $Cck^{Cre}$ mice but not in CCK-deficient mice (**Figure 2F and G** lower panel, red and blue traces, averaged ΔF/F0%: two-way ANOVA with Bonferroni multiple comparisons adjustment, significant interaction F[1, 39]=12.70, p=0.001; before vs. after HFLS in $Cck^{Cre}$: −0.003±0.017 vs 0.396±0.095, p<0.001, n=21, N=11; before vs. after HFLS in CCK-deficient: 0.000±0.018 to -0.031±0.098, ns, p=0.723, n=20, N=10; After HFLS in $Cck^{Cre}$ vs. CCK-deficient: 0.396±0.095 to -0.031±0.098, p=0.003). These results show that CCK is released from thalamocortical terminals in response to high-frequency activation.

In the previous experiment, the use of two different viruses in $Cck^{Cre}$ and CCK-deficient mice raised potential limitations regarding the suitability of this control. Additionally, the widespread distribution of CCK in the brain suggests that CCK may be released from other projections via indirect mechanisms, especially given the polysynaptic responses that can be elicited in ACx by MGB stimulation. Furthermore, laser stimulation of the thalamocortical projection may induce antidromic activation in the MGB. To address these concerns, we specifically downregulated CCK expression in MGB neurons and subsequently assessed CCK release after HFS of the MGB. We separately injected the CCK-sensor virus into the ACx and the anti-$Cck$/anti-Scramble shRNA virus into the MGB of C57 mice (**Figure 2H** upper). After viral expression was achieved (**Figure 2H** lower panel), HFS was applied to the MGB, and changes in fluorescence intensity were monitored in the ACx (**Figure 2I**). In the anti-Scramble group, HFS of the MGB induced significant fluorescence increases in the ACx (**Figure 2J**, gray, before vs. after HFS: −0.005±0.013 vs 0.387±0.064, p<0.001, n=21 from 11 mice). However, no significant fluorescence changes were observed in the anti-$Cck$ group (blue, before vs. after HFS in anti-$Cck$: −0.007±0.013 vs -0.007±0.063, p=0.999, n=22 from 11 mice). Fluorescence intensity after HFS was significantly greater in the anti-Scramble group compared to the anti-$Cck$ group (**Figure 2I** lower panel and 2 J, two-way ANOVA with Bonferroni multiple comparisons adjustment, significant interaction F[1,41]=19.70, p<0.001; anti-Scramble vs. anti-$Cck$ after HFS: 0.387±0.064 to -0.007±0.063, p<0.001). These results demonstrate that CCK release in the ACx, induced by HFS of the MGB, originates from MGB neurons and depends on intact CCK expression.

Collectively, HFS of the MGB facilitates thalamocortical synaptic plasticity through homosynaptic CCK release.

## CCK and CCKBR expression correlate with developmental thalamocortical LTP

The mechanism of LTP underlying thalamocortical plasticity in adult mice depends on CCK, which differs from that observed in neonatal brains (*Crair and Malenka, 1995*; *Barkat et al., 2011*). To investigate the emergence of CCK-dependent thalamocortical plasticity and whether age-dependent differences in HFS-induced LTP are associated with $Cck$ or CCKBR expression, we analyzed $Cck$ mRNA levels in the MGB and the distribution of CCKBR in the ACx across different age groups. Using RNAscope in situ hybridization, we examined $Cck$ mRNA expression in the MGB of juvenile (P14 and P20), young adult (8 W), and aged (18 M) mice (**Figure 3A**). At P14, only a small subset of MGB neurons expressed $Cck$ mRNA. By P20, the number of $Cck$ mRNA-expressing neurons increased significantly, followed by a slight decline at 8 W and a marked reduction at 18 M (**Figure 3B** upper panel, One-way ANOVA with Bonferroni multiple comparisons adjustment: P14 vs. P20, p<0.001; P20 vs. 8 W, p=1.0; 8 W vs. 18 M, p=0.018. P14, 57.0±41.5, n=4 from 2 mice; P20, 276.0±34.2, n=4 from 2 mice; 8 W, 232.3±33.4, n=4 from 2 mice; 18 M, 100±26.9, n=4 from 2 mice; The definition of '$Cck$ mRNA-expressing neurons' is provided in the Methods section). In contrast, no $Cck$ mRNA-expressing neurons were detected in the MGB of young adult (8 W) CCK-deficient mice (CCK-deficient, 0.0±0.0, n=8 from 4 mice; 18 Mvs. CCK-deficient, p = 0.05; 8 Wvs. CCK-deficient, p<0.001). To further assess expression dynamics, we quantified the normalized $Cck$ mRNA intensity per neuron (**Figure 3B**, lower panel). The intensity

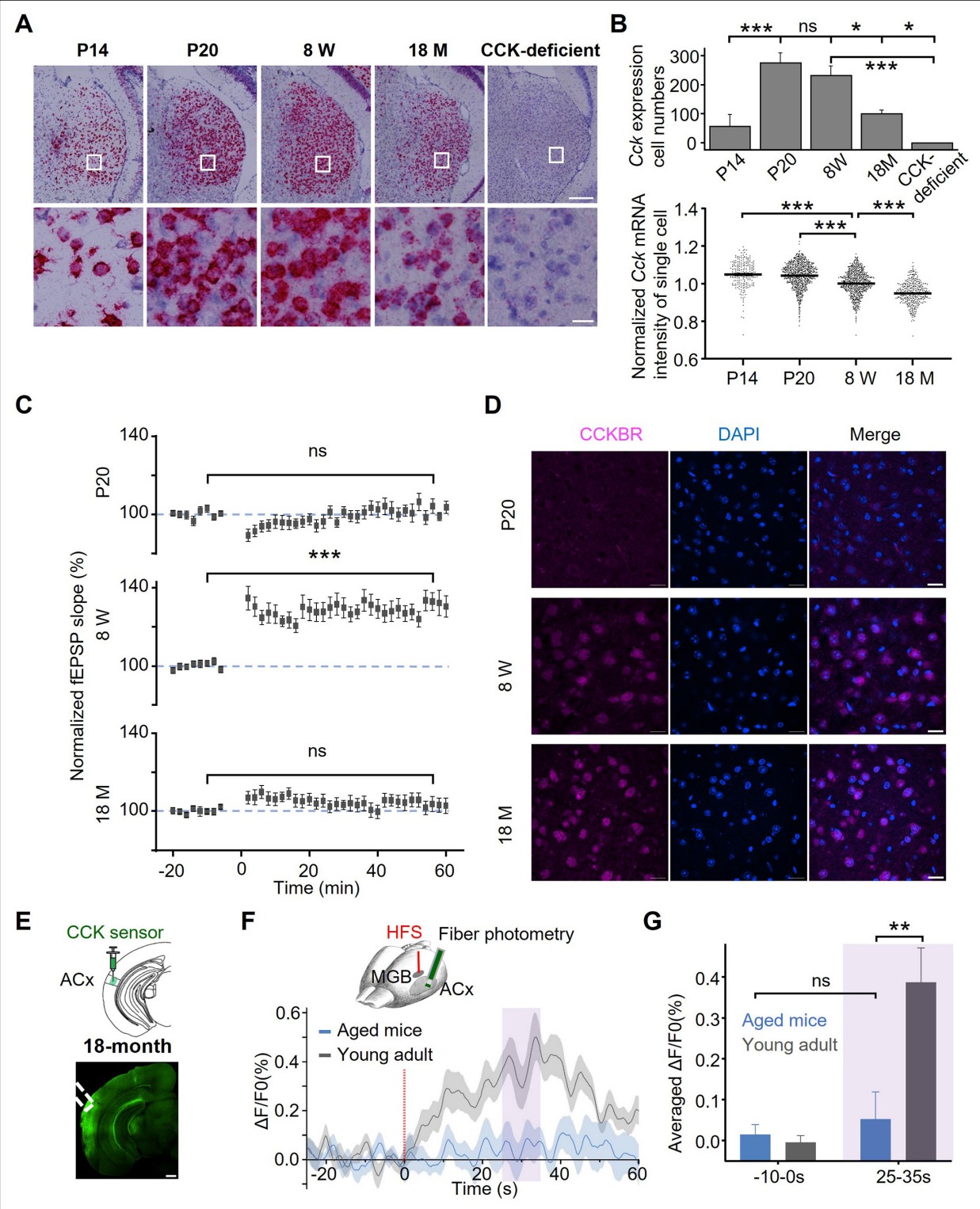

**Figure 3.** Cholecystokinin correlates with developmental thalamocortical LTP in the MGB. (**A**) Expression of *Cck* mRNA in the MGB of different age groups and in CCK-deficient mice. Scale bars: Upper, 200 μm; Lower, 20 μm. (**B**) Quantification of *Cck* expression in the MGB. Upper panel: The total number of *Cck*-expressing cells in the MGv (see Methods) for each group (One-way ANOVA, Bonferroni multiple comparisons adjustment: P14 vs. P20, ***, p<0.001; P20 vs. 8 W, ns, p=1.0; 8 W vs. 18 M, *, p=0.018; 18 M vs. CCK-deficient, *, p=0.05; 8 Wvs. CCK-deficient, ***, p<0.001. n of P14=4 from2 mice; n of P20=4 from2 mice; n of 8W=4 from2 mice; n of 18M=4 from2 mice; n of CCK-deficient = 8 from4 mice). Lower panel: Normalized *Cck* mRNA intensity per neuron (see Methods) across developmental stages (One-way ANOVA, Bonferroni multiple comparisons adjustment: P14 vs. 8 W, ***, p<0.001; P20 vs. 8 W, ***, p<0.001; 8 W vs. 18 M, ***, p<0.001; n of P14=228 neurons; n of P20=1104 neurons; n of 8W=929 neurons;

*Figure 3 continued on next page*

*Figure 3 continued*

n of 18M=400 neurons). (**C**) HFS-induced thalamocortical LTP in different age groups. Normalized fEPSP slopes before and after HFS in the MGB at P20 (upper, ns, p=0.542, n=15 from 10 mice), 8 W (middle, ***, p<0.001, n=13 from 10 mice), and 18 M(lower, ns, p=0.380, n=18 from 8 mice). (**D**) Immunostaining of CCKBR in the ACx neurons at different ages (P20, 8 W, and 18 M). CCK: red; DAPI: blue. Scale bars: 20 µm. (**E**) Upper: Injection of AAV9-syn-CCKsensor into ACx of aged mice (18 M). Lower: Enlarged images displaying the CCK-sensor expression in the ACx. Scale bar: 500 µm. (**F**) Upper: Experiment design for HFS application in the MGB and fluorescence monitoring in the ACx. Lower: Traces of fluorescence signals of the CCK-sensor before and after HFS in aged mice (blue). Fluorescence signals from the anti-Scramble group (gray, adult control, 3–4 months) are included for comparison. The averaged ΔF/F0% is shown as solid lines, with the shaded areas representing SEM. (**G**) Bar charts showing the averaged ΔF/F0% before and after HFS for aged and adult control mice (Statistical analysis with Bonferroni multiple comparisons adjustment: before vs. after HFS in aged mice: ns, p=0.699, n=12 from 6 mice; after HFS in aged mice vs. adult control group: **, p=0.01).

peaked during postnatal development at the juvenile stage, decreased modestly in young adulthood, and declined substantially in aged mice (*Figure 3B* lower panel, One-way ANOVA with Bonferroni multiple comparisons: P14 vs. 8 W, p<0.001; P20 vs. 8 W, p<0.001; 18 M vs. 8 W, p<0.001; P14, 1.048±0.004, n=228 neurons; P20, 1.042±0.002, n=1104 neurons; 8 W, 1.000±0.002, n=929 neurons; 18 M, 0.949±0.003, n=400 neurons). Overall, these results indicate that *Cck* mRNA expression in the MGB is low at P14, reaches its highest level at P20, decreases slightly at 8 W, and drops dramatically at 18 M (*Figure 3B*).

To investigate the correlation between HFS-induced thalamocortical LTP and *Cck* expression, we conducted in vivo electrophysiology experiments in age groups P20, 8 W, and 18 M mice, while excluding P14 mice due to their small body size (*Figure 3C*). As expected, HFS of MGB induced robust LTP in the 8 W mice (*Figure 3C* middle panel, data are shown as in *Figure 1D*, indicating an increase of 30.0 ± 4.8%, p<0.001). Similarly, the absence of LTP in 18 M mice was consistent with their low *Cck* mRNA expression (*Figure 3C* lower panel, an increase of 2.9 ± 3.2%, p=0.380). However, no LTP was observed in P20 mice, despite their high *Cck* mRNA levels (*Figure 3C* upper panel, increased by 2.2 ± 3.5%, p=0.542; *Figure 3C*, two-way RM ANOVA, F[2,42]=18.7, p<0.001, pairwise comparison: before vs. after HFS in P20 mice, 100.0±0.5% vs 102.2 ± 3.6%, p=0.542, n=15 from 10 mice; before vs. after HFS in 8 W mice, 100.7±0.4% vs 130.7 ± 4.8%, p<0.001, n=13 from 10 mice; before vs. after HFS in 18 M mice, 100.5±0.4% vs 103.3 ± 3.3%, p=0.380, n=18 from 8 mice). This unexpected result suggests that while *Cck* expression is critical, it is not sufficient on its own to support HFS-induced thalamocortical LTP. Although LTP levels decreased with reduced *Cck* mRNA expression in mature mice (8 W and 18 M; difference between 8 W and 18 M after HFS: 27.4 ± 5.1%, p<0.001), the lack of LTP in P20 mice (difference between 8 W and P20 after HFS: 28.5 ± 5.3%, p<0.001) led us to speculate that the differential thalamocortical plasticity mechanisms between neonatal and adult mice might be attributed to differences in the expression of CCKBR. The absence of CCKBR in P20 mice could prevent CCK from exerting its effects, thereby impairing HFS-induced LTP. To test this hypothesis, we examined CCKBR expression levels in the ACx across developmental stages. As predicted, immunochemistry staining revealed minimal CCKBR signals in P20 mice, whereas strong signals were observed in the 8 W and 18 M groups (*Figure 3D*). These results indicate that the maturation of thalamocortical projections during development is accompanied by an increase in CCKBR expression. In conclusion, HFS-induced thalamocortical LTP correlated with *Cck* expression in adult mice but not in neonatal brain, likely due to the limited expression of CCKBR during the critical period.

However, a concern arises regarding our theory. Although the expression of *Cck* mRNA in aged mice is significantly reduced, it is not entirely absent (*Figure 3B* right panel). In contrast, thalamocortical LTP was barely inducible in aged mice (*Figure 3C* lower panel), suggesting the lack of CCK release, rather than the absence of *Cck* mRNA, may underlie this deficit. It is widely recognized that mRNA levels do not always directly correlate with peptide levels due to multiple steps involved in peptide synthesis and processing, including translation, post-translational modifications, packaging, transportation, and proteolytic cleavage, all of which require various enzymes and regulatory mechanisms (*Mierke, 2020*; *Gualillo et al., 2006*; *Sossin et al., 1989*; *Hook et al., 2008*). A disruption at any stage in this process could lead to impaired CCK release, even when *Cck* mRNA is present. To directly assess whether CCK release is impaired in aged mice, we utilized the CCK sensor to measure fluorescence changes in the ACx during HFS of the MGB. In aged mice (18 M), no significant increase in fluorescence intensity was observed following HFS (*Figure 3F*, blue), in stark contrast to the robust fluorescence increase seen in the adult control group (3–4 months, *Figure 3F*, gray,

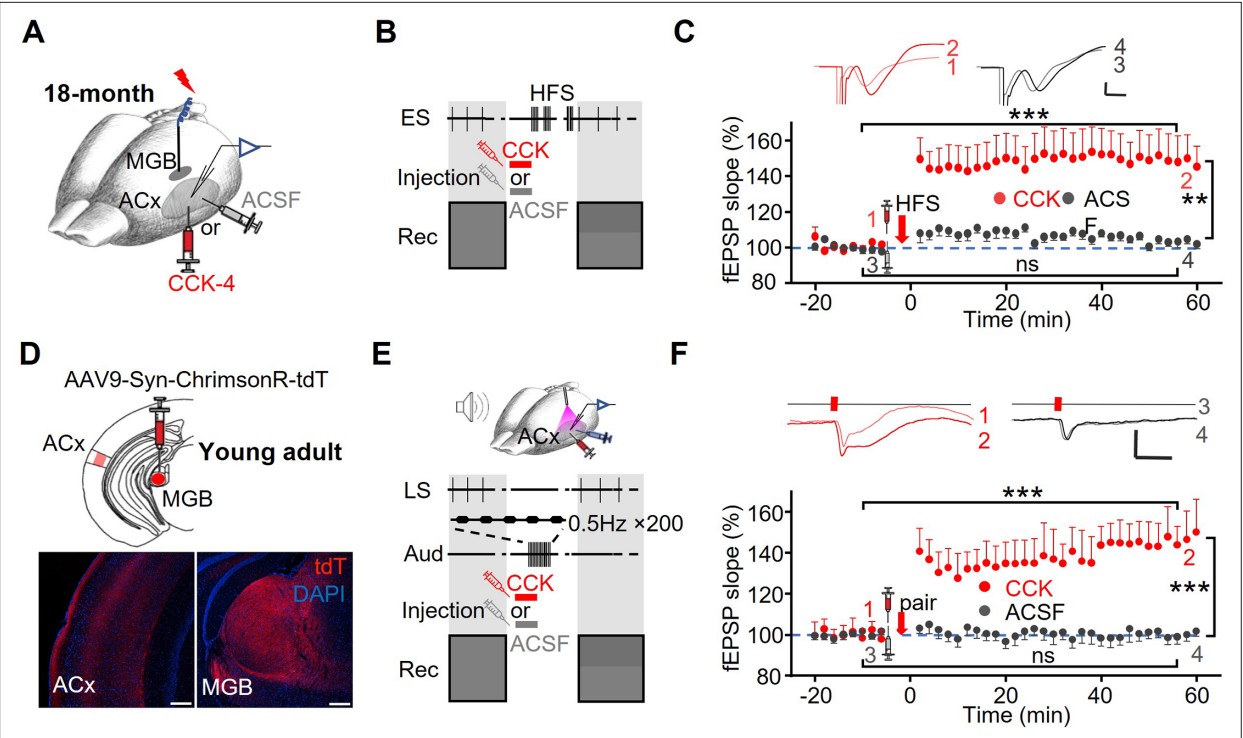

**Figure 4.** Exogenous application of CCK rescues thalamocortical connectivity. (**A**) Schematic representation of the experimental setup, showing electrode placement and drug injection sites. (**B**) Experimental protocol. CCK or ACSF was injected into the ACx, followed by HFS in the MGB. fEPSPs in response to MGB ES were recorded before and after the intervention. (**C**) CCK-4 injection restored HFS-induced thalamocortical LTP in aged mice. Upper: Representative traces of fEPSPs recorded before (1,3) and after (2,4) drug injection followed by HFS in the CCK-4 (red) and ACSF (gray) groups. Scale bar: 10ms, 0.1 mV. Lower panel: Normalized fEPSP slopes before and after CCK-4 (red, ***, p<0.001, n=13 from 6 mice) or ACSF (gray, ns, p=0.404, n=12 from 7 mice) injection and subsequent HFS in aged mice (two-way ANOVA, CCK vs. ACSF after the intervention: **, p=0.006). (**D**) Upper: AAV9-hSyn-ChrimsonR-tdTomato was injected into the MGB of C57 mice. Lower panel: Histological confirmation of viral expression in the ACx and MGB. Scale bars: 100 μm. (**E**) Upper panel: Schematic Diagram of the Experimental Setup. A 620 nm laser was used to activate thalamocortical terminals expressing opsins in the ACx. Glass pipette electrodes in the ACx recorded field EPSPs evoked by laser stimulation. Lower panel: Experimental protocol. CCK or ACSF was injected into the ACx, followed by 200 auditory stimulations. fEPSPs in response to laser stimulation were recorded before and after the intervention. (**F**) Upper: Representative fEPSP traces evoked by laser before (1,3) and after (2,4) the intervention (drug injection followed by auditory stimulation). Scale bars: 40ms, 0.1 mV. Lower: Normalized slopes of laser-evoked fEPSPs for 16 mins before and 1 hr after the intervention in the CCK-4 and ACSF groups (two-way RM ANOVA, before vs. after: CCK group, ***, p<0.001, n=10 from 10 mice; ACSF group, ns, p=0.899, n=19 from 10 mice; CCK vs. ACSF after the intervention, p<0.001).

from *Figure 2I*, anti-Scramble group; *Figure 3G*, Averaged ΔF/F0%: two-way ANOVA with Bonferroni multiple comparisons adjustment, significant interaction F[1,31]=8.61, p=0.006; before vs. after HFS in aged mice: 0.014±0.022 vs 0.052±0.098, p=0.699, n=12 from 6 mice; After HFS in aged mice vs. anti-Scramble control: 0.052±0.098 vs 0.387±0.074, p=0.010). These findings confirm that HFS of the MGB fails to trigger CCK release in the aged brain, despite the presence of *Cck* mRNA. A shortage of CCK may contribute to the impairment of thalamocortical plasticity in aged mice.

## Exogenous CCK application restores thalamocortical LTP in aged mice and enhances frequency discrimination

Previous studies have shown that age-related impairment in synaptic plasticity, including LTP, contributes to hippocampal dysfunction (*Deupree et al., 1993*), and that synaptic plasticity declines with age in the thalamocortical pathway of rats (*Dahmen and King, 2007*; *Hogsden and Dringenberg, 2009*; *Speechley et al., 2007*). Our findings revealed that thalamocortical LTP cannot be induced in aged mice, likely due to insufficient CCK release, despite intact CCKBR expression. Given these observations, we hypothesized that exogenous CCK application could compensate for the deficit in endogenous CCK release and potentially restore thalamocortical LTP in aged mice. To test this, we infused CCK-4 (0.5 μL, 10 μM) directly into the ACx of aged mice (18 M) prior to HFS at the MGB

(*Figure 4A and B*). Remarkably, this intervention restored thalamocortical LTP, as evidenced by a significant increase in fEPSP slopes after HFS compared to baseline (*Figure 4C*, two-way RM ANOVA, F[1,22]=17.9, p<0.001, pairwise comparison: CCK group, before vs. after, 100.4±0.6% vs 148.6 ± 10.4%, p<0.001, n=13 from 6 mice). In contrast, no successful LTP induction was observed in the ACSF control group (ACSF group, before vs. after, 98.8±0.7% vs 103.3 ± 4.9%, p=0.404, n=12 from 7 mice; The difference between the CCK and ACSF groups after the intervention was 45.3 ± 15.1%, p=0.006), further supporting the specific role of CCK in rescuing thalamocortical plasticity. Based on previous studies, CCK facilitates synaptic plasticity through presynaptic and postsynaptic coactivation (*Li et al., 2014*; *Chen et al., 2019a*). To investigate whether CCK alone is sufficient to induce thalamocortical LTP without activating thalamocortical projections, we infused CCK-4 into the ACx of young adult mice immediately after baseline fEPSPs recording. Stimulation was then paused for 15 min to allow for CCK degradation, after which recording resumed. Notably, CCK infusion alone failed to induce LTP (*Figure 5—figure supplement 1A*, one-way RM ANOVA, F[1,19]=0.003; pairwise comparison, before vs. after CCK alone, 99.1±0.4% vs 99.2 ± 1.4%, p=0.956, n=20 from 5 mice), indicating that activation of the thalamocortical pathway is essential for LTP induction, while CCK serves as a key regulator of this process.

To explore the potential therapeutic application of CCK, we hypothesized that combining CCK administration with natural auditory stimulation, which engages thalamocortical circuits, might synergistically induce plasticity. This approach addresses the need for non-invasive methods to enhance neural plasticity, as direct MGB stimulation is not clinically feasible. First, we labeled thalamocortical projections in the ACx by injecting AAV9-Syn-ChrimsonR-tdTomato into the MGB of C57 mice (*Figure 4D*). In electrophysiological experiments, either CCK-4 or ACSF was infused into the ACx, followed by 200 trials (0.5 Hz) of auditory stimulation (*Figure 4E*). Laser-evoked fEPSPs were recorded before and after the intervention. Thalamocortical LTP was successfully induced in the CCK group, while no significant changes were observed in the ACSF control group (*Figure 4F*, two-way RM ANOVA, F[1,27]=30.7, p0.001, pairwise comparison: CCK group, before vs. after, 99.9±0.9% vs 146.1 ± 7.0%, p<0.001, n=10 from 10 mice; ACSF group, before vs. after, 100.7±0.7% vs 100.0 ± 5.1%, p=0.899, n=19 from 10 mice; The difference between CCK group and ACSF group after the intervention was 46.1 ± 8.6%, p<0.001). These results demonstrate that the combination of CCK infusion and auditory stimulation effectively induces thalamocortical plasticity. To confirm whether auditory stimulation is necessary for inducing thalamocortical LTP, we infused CCK-4 into the ACx, paused auditory stimulation, and suspended the recording of laser-evoked fEPSPs until CCK degradation. In this condition, no significant potentiation was observed (*Figure 5—figure supplement 1B*, one-way RM ANOVA, F[1,21]=0.546; pairwise comparison, before vs. after CCK alone, 100.3±0.4% vs 99.0 ± 1.9%, p=0.468, n=22 from 4 mice). Collectively, these findings indicate that while CCK is essential for modulating thalamocortical plasticity, auditory stimulation is required to activate this pathway and induce LTP.

Auditory thalamocortical plasticity plays a decisive role in inducing precise modifications of cortical neurons to support frequency-specific plasticity in the auditory cortex (*Liu et al., 2011*; *Jafari et al., 2007*). Sensory experiences, such as sound exposure, are known to influence thalamocortical plasticity in adulthood (*Oberlaender et al., 2012*; *Zhou et al., 2011*; *Speechley et al., 2007*). Chen et al. demonstrated that optogenetic silencing of the auditory thalamocortical projection impairs auditory decision-making during frequency-discrimination tasks (*Chen et al., 2019b*). In our subsequent experiment, we used a prepulse inhibition (PPI) acoustic startle test to evaluate whether CCK administration could improve frequency discrimination ability in mice (*Figure 5A*). PPI reflects the animals' perception of frequency differences between the background tone and the prepulse tone, with higher PPI values indicating better discrimination ability (*Figure 5B*). After habituation, young adult mice were exposed to tone stimulations (9.8 or 16.4 kHz, used as background tones in the PPI test) in a soundproof chamber immediately following bilateral injection of either CCK or ACSF into the ACx through implanted cannulas (*Figure 5A*). Frequency discrimination ability was assessed by comparing PPIs between the two groups 24 hours after exposure. As expected, in both ACSF control groups (*Figure 5C*, 9.8 kHz; *Figure 5D*, 16.4 kHz, gray lines), PPI values gradually increased as the frequency difference (Δf) between the background and prepulse tones widened. Notably, the CCK-treated groups exhibited significantly higher PPI values than the ACSF groups when detecting prepulse tones near the exposed frequency. Specifically, in both the 9.8 kHz (*Figure 5C*) and 16.4 kHz (*Figure 5D*)

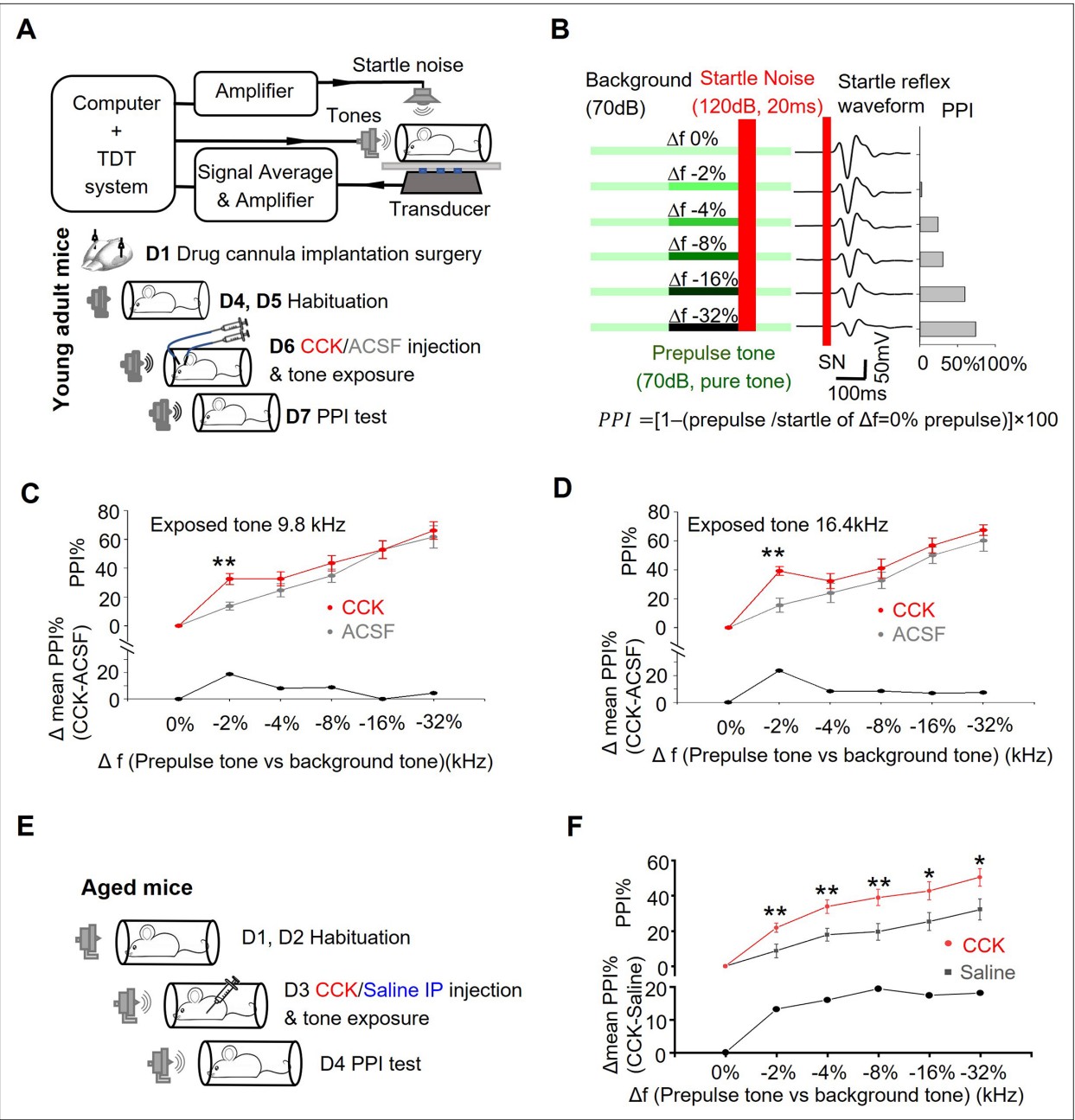

**Figure 5.** Exogenous application of CCK enhances frequency discrimination. (**A**) Upper: Schematic diagram of the experimental setup for the PPI test. lower: Experimental timeline showing tone exposure and CCK/ACSF injection into the ACx of young adult mice. (**B**) Left: Schematic representation of the PPI acoustic startle test protocol. A continuous background pure tone (70 dB, 9.8 kHz, or 16.4 kHz) was presented during the test, except when prepulse tones and startle noise bursts were delivered. PPI trials were presented in a pseudorandom order. Each prepulse test consisted of an 80ms prepulse at 70 dB (Δf, pure-tone frequency was 0%, 2%, 4%, 8%, 16%, or 32% lower than the background tone), followed by a 20ms white noise startle pulse at 120 dB, after which the background tone resumed. Each trial was repeated 15 times. Middle: Averaged startle waveforms from a representative mouse. Right: Calculated PPI (as defined below) for the same example. (**C**) Mean PPI (%) of the startle responses in young adult mice exposed to a 9.8 kHz tone after CCK-8S (red) or ACSF (gray) injection into the ACx (CCK, n=7, ACSF n=7, two-way ANOVA, p<0.05, post-hoc, Tukey test, CCK vs. ACSF at Δf = –2%, **, p<0.001). Prepulse frequencies were 0%, 2%, 4%, 8%, 16%, or 32% lower than the background tone, 9.8 kHz. The difference in the mean PPI between the CCK infusion and ACSF infusion groups is listed as a function of the Δf in the lower panel. (**D**) Mean PPI (%) of startle responses in young adult mice exposed to a 16.4 kHz tone after CCK-8S (red) or ACSF (gray) injection into the ACx (CCK n=8, ACSF n=7, two-way ANOVA, p<0.05, post-hoc, Tukey test, CCK vs. ACSF at Δf = –2%, **, p<0.001). Prepulse frequencies were 0%, 2%, 4%, 8%, 16%, or 32% lower than the background tone, 16.4 kHz. The difference in the mean PPI between the CCK infusion and ACSF infusion groups is listed as a function of the Δf in the lower panel. (**E**) Experimental timeline for tone exposure and CCK/saline i.p. injection in aged mice. (**F**) Mean PPI (%) of startle responses in aged mice exposed to a 9.8 kHz tone after i.p. injection of CCK-4 (red) or saline (gray). Prepulse frequencies were 0%, 2%, 4%, 8%, 16%, or 32% lower than the background

*Figure 5 continued on next page*

*Figure 5 continued*

tone, 9.8 kHz. The difference in mean PPI between the CCK-4 injection and saline injection groups is listed as a function of the Δf in the lower panel (Bonferroni-adjusted multiple comparisons with CCK vs. saline at Δf = : **–2%**, **, p=0.010; **–4%**, **, p=0.006; **–8%**, **, p=0.006; **–16%**, *, p=0.022; **–32%**, *, p=0.027; Frequency-discrimination task in CCK-4 group vs. saline group, two-way ANOVA, **, p=0.004; CCK-4, N=15; saline, N=15).

The online version of this article includes the following figure supplement(s) for figure 5:

**Figure supplement 1.** Exogenous CCK restores thalamocortical LTP in aged mice and improves frequency discrimination ability.

tone exposure groups, CCK-treated mice showed a>18.8% improvement in PPI compared to ACSF-treated mice at Δf = –2% (*Figure 5C*, 9.8 kHz, CCK group, n=7, ACSF group, n=7, two-way ANOVA, p<0.05, 32.50 ± 3.86% vs 13.67 ± 2.77%, post-hoc, Tukey test, CCK vs. ACSF, **, p<0.001; *Figure 5D*, 16.4 kHz, CCK group, n=7, ACSF group, n=7, two-way ANOVA, p<0.05, 39.15 ± 3.01% vs 15.43 ± 4.97%, post-hoc, Tukey test, CCK vs. ACSF, **, p<0.001). Interestingly, the improvement in frequency discrimination in young adult mice was limited to prepulse tones close to the exposed frequency, with no significant differences observed at larger Δf values (e.g. Δf = –8%, –16%, or –32%). This plateau in performance may reflect the already efficient thalamocortical connectivity in young animals, leaving little room for enhancement through exogenous CCK application. This phenomenon underscores the role of the auditory thalamocortical system in mediating precise, frequency-specific plasticity in cortical function (*Liu et al., 2011*; *Jafari et al., 2007*).

Supporting this result, a pilot study conducted in rats (*Figure 5—figure supplement 1C–E*) demonstrated similar effects. In anesthetized rats, we infused CCK-8S into the ACx and delivered tone stimuli for 200 trials, measuring the auditory cortical neuron tuning curves before and after the intervention (*Figure 5—figure supplement 1C*). The frequency of the selected exposed tone (EF) was a non-characteristic frequency that elicited a moderate response. Although CCK infusion did not alter the tuning curve's characteristic frequency (CF), it significantly lowered the response threshold to the exposed tone. In contrast, ACSF infusion did not affect the response threshold (*Figure 5—figure supplement 1D* for an example; *Figure 5—figure supplement 1E* for group data, 8.66±1.91 dB at exposed frequency, p<0.001; 6.66±2.32 dB p<0.001 and 5.33±1.33 dB, p<0.05 at frequencies of 0.4 octave lower and 0.4 octave higher than the exposed frequency, respectively; CCK, n=15; ACSF n=12, two-way ANOVA, p<0.001, post hoc: Tukey test, CCK-8S vs. ACSF, *, p<0.05; **, p<0.001). Together, these results indicate that CCK combined with sound exposure enhances cortical sensitivity to the exposed frequency range, which may explain the observed improvements specifically in discriminating the –2% difference during behavioral tests.

Age-related atrophy in thalamocortical connectivity contributes to cognitive decline (*Hughes et al., 2012*). In our study, we identified deficits in CCK-dependent thalamocortical plasticity in aged mice and successfully restored this plasticity by exogenous CCK administration into the ACx. To further explore its therapeutic potential, we examined whether intraperitoneal (i.p.) administration of CCK could enhance frequency discrimination ability in aged mice, as i.p. injection is a more feasible delivery method than intracortical administration. After habituation, 15 aged mice (18 M) received i.p. injections of CCK-4 solution (1 mg/kg), while 15 littermate controls received saline injections. Fiber photometry in CCK-sensor-expressing mice confirmed that CCK-4 crosses the blood-brain barrier and reaches the ACx (*Figure 5—figure supplement 1F–H*, Averaged ΔF/F0% of *Cck*[Cre] mice: two-way ANOVA with Bonferroni multiple comparisons adjustment, F[1,14]=7.465, p=0.016, before vs. after CCK-4 treatment: 0.003±0.036 vs 0.523±0.154, p=0.002, N=7; before vs. after saline treatment: 0.003±0.032 vs 0.011±0.135, p=0.951, N=9; After CCK-4 vs. after saline: 0.523±0.154 vs 0.011±0.135, p=0.025; Averaged ΔF/F0% of CCK-deficient mice: two-way ANOVA with Bonferroni multiple comparisons adjustment, F[1,13]=22.99, p<0.001; before vs. after CCK-4 treatment: 0.024±0.037 vs 0.716±0.122, p<0.001, N=7; before vs. after saline treatment: 0.025±0.035 vs 0.009±0.115, p=0.877, N=8; After CCK-4 vs. after saline: 0.716±0.122 vs 0.009±0.115, p=0.001). Immediately after injection, mice were exposed to tone stimuli (9.8 kHz) in a soundproof chamber (*Figure 5E*). The PPI testing 24 hr later revealed significant improvements in frequency discrimination in the CCK group compared to the saline group across almost all tested prepulse frequencies (*Figure 5F*, two-way ANOVA, test of between-drug effects in frequency-discrimination task: F[1,28]=9.572, p=0.004, CCK group N=15, saline group N=15. Bonferroni-adjusted pairwise multiple comparisons with CCK vs. saline in Δf = **–2%**: 21.8±3.3% vs 8.7 ± 3.3%, p=0.010; Δf = **–4%**: 33.8±3.8% vs 17.8 ± 3.8%, p=0.006; Δf = **–8%**: 38.9±4.6% vs 19.5 ± 4.6%, p=0.006; Δf = **–16%**: 42.7±5.1% vs 25.3 ± 5.1%, p=0.022; Δf = **–32%**: 50.4±5.5% vs 32.3 ±

5.5%, p=0.027). The differences in CCK efficacy between young adults (*Figure 5C and D*) and aged animals may stem from variations in endogenous CCK expression across age groups. Overall, these findings suggest that combining CCK administration with sound stimulation holds promise as a potential therapeutic approach for mitigating cognitive decline associated with auditory thalamocortical connectivity impairments or deficits in thalamocortical plasticity.

## Discussion

In this study, we demonstrated that CCK is a critical modulator of thalamocortical LTP in the ACx, with its role varying across developmental stages and aging. Our results revealed that HFS of the MGB successfully induced thalamocortical LTP in young adult mice, and this induction was critically dependent on CCK, as evidenced by the loss of LTP following *Cck* knockdown or pharmacological blockade of CCKBR in the ACx. The reemergence of thalamocortical LTP beyond the neonatal critical period was associated with the developmental upregulation of CCK and CCKBR, while the loss of thalamocortical LTP in aged mice was likely due to reduced CCK expression and impaired CCK release in the thalamocortical pathway. Notably, exogenous CCK application into the ACx restored thalamocortical LTP in aged mice, and when combined with auditory exposure, significantly enhanced thalamocortical connectivity and improved frequency discrimination, highlighting its potential as a foundation for exploring therapeutic approaches targeting neuroplasticity deficits.

### Thalamocortical long-term potentiation

Thalamocortical plasticity plays a crucial role in the development of sensory cortices. Previous studies have shown that the refinement of thalamocortical projections during early development is driven by activity-dependent mechanisms, including competition among thalamic axons for cortical targets (*Barkat et al., 2011*, *Goodman and Shatz, 1993*). The switching of NMDA receptor subunits (NR2B to NR2A) at thalamocortical synapses during early development marks a critical period for NMDA-dependent LTP at these synapses (*Liu et al., 2004*). Correlated pre- and postsynaptic activity contributes to the conversion of silent thalamocortical synapses into functional synapses, which is essential for neonatal thalamocortical development (*Isaac et al., 1997*). Zhang et al. showed that exposing neonatal rats to pulsed monotone stimuli from P9 to P28 resulted in an expansion of the cortical area representing the exposed tones in the ACx (*Zhang et al., 2001*). These developmental changes are mediated by NMDA receptors, and this plasticity declines as NMDA receptor subunits switch from NR2B to NR2A during the critical period (*Liu et al., 2004*; *Isaac et al., 1997*), indicating reduced sensitivity to passive sensory inputs and affecting the gating of thalamocortical plasticity.

Although thalamocortical plasticity is most prominent during early development, studies have demonstrated its persistence in the adult brain under specific induction protocols (*Blundon et al., 2017*; *Hogsden and Dringenberg, 2009*; *Heynen and Bear, 2001*). Consistent with *Heynen and Bear, 2001*, our findings show that thalamocortical LTP can be induced in vivo using the standard HFS protocol, indicating that thalamocortical synapses retain their plasticity beyond the early critical period as defined by Malenka (*Crair and Malenka, 1995*). This LTP induction enhanced auditory signal propagation, providing strong support of the functional impact of HFS-induced thalamocortical plasticity (*Figure 1*).

### CCK dependence of thalamocortical neuroplasticity in the ACx

Thalamocortical LTP in the ACx exhibits a lower dependence on NMDARs compared to the somatosensory or visual cortices (*Chun et al., 2013*). Auditory thalamocortical plasticity becomes gated after the early critical period, relying on distinct signaling mechanisms that differ from those in other sensory cortices (*Chun et al., 2013*; *King and Nelken, 2009*). Previous studies have revealed the crucial role of CCK in facilitating cortical plasticity in adult rats and mice (*Li et al., 2014*; *Chen et al., 2019a*; *Li et al., 2023*; *Zhang et al., 2020*). Additionally, Senatorov et al. reported the presence of *Cck* mRNA in the reciprocally connected areas of the MGB and the ACx, with a high density of CCKBR in layer IV of the ACx (*Zarbin et al., 1983*; *Senatorov et al., 1997*). These findings suggest a potential involvement of CCK in thalamocortical plasticity. Our data extend this framework by identifying CCK–CCKBR signaling as a permissive modulator of adult thalamocortical LTP. Specifically, high-frequency activation of thalamocortical pathway triggered CCK release, which facilitated

thalamocortical LTP (*Figure 2B and F*). Knocking down CCK expression in MGB neurons abolished HFS-induced CCK release and subsequent LTP (*Figure 2D and I*). Similarly, pharmacological blockade of CCKBR in the auditory cortex effectively impaired LTP induction in wild-type mice (*Figure 2—figure supplement 1D*). These results establish CCK as an indispensable component of thalamo-cortical neuroplasticity. We propose that CCKBR activation may trigger intracellular calcium release and AMPAR recruitment in parallel to, or partially compensating for, postsynaptic NMDAR signaling (*Chen et al., 2019a*; *Li et al., 2023*). This complementary arrangement may reconcile differences across developmental stages and cortical areas and highlights neuropeptidergic signaling as a lever to re-enable adult thalamocortical plasticity.

Notably, exogenous CCK alone failed to induce LTP in the absence of accompanying stimulation (*Figure 5—figure supplement 1A and B*), emphasizing that CCK functions as a modulator rather than a direct initiator of LTP. Activation of the thalamocortical pathway is also essential for LTP induc-tion. Although our experiment targeting the MGv was guided by stereotaxic coordinates and veri-fied post hoc, we acknowledge potential contributions from non-lemniscal medial geniculate nucleus dorsal (MGd) projections. Anatomical and physiological evidence indicates that MGv-AC projections provide rapid, frequency-specific, tonotopically organized excitation, whereas MGd pathways target higher-order auditory cortex with broader tuning, less precise tonotopy, longer response latencies, and greater context dependence, features that can differentially shape cortical sensory integration and plasticity (*Lee and Sherman, 2010*; *Smith et al., 2012*; *Ohga et al., 2018*; *Lee, 2015*; *Hu, 2003*). While the co-recruitment of lemniscal and non-lemniscal inputs may enhance the generality of our CCK-dependent mechanism, the differing response characteristics of these pathways suggest subtle differences in their relative engagement in the observed plasticity. Future pathway-specific manipu-lations will help clarify their respective contributions. Another potential limitation of our study is the trans-synaptic transfer property of AAV9 (*Figure 2—figure supplement 1F*). To mitigate this risk, we carefully control the injection volume, rate, and viral expression time, while also verifying expression post-hoc. Systematic sampling histological analysis detected no tdTomato-positive cortical somata in the ACx (*Figure 2E* lower panel), whereas rare EYFP-positive cortical somata were observed after AAV9-EF1a-DIO-ChETA-EYFP injections (median <1 cell in 0.4×0.4 mm$^2$ section, *Figure 2—figure supplement 1F*, corresponds to *Figure 2A* upper-middle panel). These construct-dependent obser-vations align with occasional low-level trans-synaptic transfer reported for AAV9 (*Zingg et al., 2017*) and indicate that off-target cortical infection was negligible for ChrimsonR and exceedingly rare for ChETA under our experimental conditions.

## Developmental and age-dependent CCK-mediated plasticity

Our findings reveal that CCK-mediated thalamocortical plasticity in the auditory system is tightly linked to specific developmental stages and undergoes significant age-related changes. In the mouse auditory system, MGB projections reach the subplate as early as P2 and activate subplate neurons (*Viswanathan et al., 2012*; *Zhao et al., 2009*), which play a crucial role in guiding thalamocortical projections to layer IV of the ACx during the first 2 postnatal weeks. As the thalamocortical connec-tions mature, the subplate neurons undergo programmed cell death around the fourth postnatal week (*Bandiera and Molnár, 2022*). These early processes are essential for the initial wiring of thalamocor-tical circuits.

However, functional thalamocortical LTP that is dependent on CCK does not emerge immedi-ately during these early stages of development (*Figure 3C*). Despite the high levels of *Cck* mRNA expression in P20 mice (*Figure 3A and B*), HFS-induced thalamocortical LTP was absent at this age (*Figure 3C*), likely due to insufficient CCKBR expression in the ACx during early postnatal develop-ment (*Figure 3D*). As CCKBR expression increases, CCK-dependent LTP is realized in adult mice (*Figure 3C and D*). In contrast, in aged mice, the absence of thalamocortical LTP correlates with significantly reduced *Cck* mRNA levels in the MGB and impaired CCK release, highlighting the critical role of CCK availability in maintaining functional plasticity in the ACx across the lifespan. This aligns with previous observations by *Hogsden and Dringenberg, 2009* that HFS-induced LTP is most prom-inent in rats around 6 weeks and diminishes with aging (*Hogsden and Dringenberg, 2009*).

Together, these findings indicate that the CCK-mediated thalamocortical plasticity exhibits devel-opmental and age-dependent characteristics, relying on both the availability of presynaptic CCK in the MGB projection and the maturation of postsynaptic CCKBR expression in the ACx.

## CCK administration restores thalamocortical LTP in aged mice and improves frequency discrimination ability

Given that aged mice exhibit reduced endogenous CCK but retain intact CCKBR, we successfully restored the impaired thalamocortical LTP by applying exogenous CCK into the ACx of aged mice (*Figure 4C*). This suggests that the absence of thalamocortical LTP in aged mice is primarily due to reduced CCK release rather than irreversible structural changes in the thalamocortical pathway. Furthermore, the combined application of CCK and acoustic stimulation enhanced thalamocortical connectivity (*Figure 4F*), highlighting its potential as a therapeutic approach. Auditory cortical plasticity is closely associated with improved perceptual acuity and learning (*Blundon et al., 2017*; *Recanzone et al., 1993*), while thalamocortical plasticity serves as a critical determinant in refining cortical neurons (*Liu et al., 2011*; *Jafari et al., 2007*). Inhibition of thalamocortical inputs has been shown to impair frequency-discrimination ability (*Chen et al., 2019b*).

In this study, frequency discrimination ability was assessed using pre-pulse inhibition (PPI) of the acoustic startle response, a behavioral paradigm that reflects sensory gating and involves multiple brain regions, with the ascending auditory pathway playing a key role (*Gómez-Nieto et al., 2020*). Within the auditory cortex, layer IV neurons receive tonotopically organized inputs from the MGB and are critical for integrating thalamic inputs and shaping auditory processing. Synaptic plasticity in layer IV, enhanced by CCK, may amplify the cortical representation of weak auditory signals, thereby improving pre-pulse detection and enhancing PPI performance. In aged mice, PPI deficits are commonly observed due to impaired auditory processing (*Ouagazzal et al., 2006*; *Young et al., 2010*). Notably, C57BL/6 mice exhibit age-related hearing loss (*Johnson et al., 1997*). Both age-associated changes in auditory function and CCK deficiency contribute to impaired sensory gating. The presence of partial hearing loss in aged mice may have facilitated the detection of CCK's beneficial effects, further highlighting its therapeutic potential for age-related deficits. Our results suggest that enhanced thalamocortical plasticity mediated by CCK might partially compensate for these deficits by amplifying residual auditory signals in aged mice.

Our behavioral experiments demonstrated that the combination of passive tone stimulation and bilateral CCK infusion into the auditory cortices of young adult mice significantly enhanced their frequency discrimination ability at the exposed frequency (*Figure 5C and D*). In comparison, passive tone stimulation combined with i.p. CCK administration in aged mice improved frequency discrimination across all tested frequencies (*Figure 5F*), likely by enhancing diminished thalamocortical communication. Notably, the effects of CCK were more pronounced in aged mice compared to young adults. It is plausible that endogenous CCK levels in young adult animals are sufficient to regulate thalamocortical plasticity, allowing them to adapt to environmental changes. In aged mice, however, the lack of sufficient endogenous CCK may explain their heightened sensitivity to exogenous CCK administration. This increased sensitivity was reflected by slightly higher fluorescence signals in CCK-deficient mice compared to $Cck^{Cre}$ mice after CCK injection (*Figure 5—figure supplement 1G*, CCK-deficient is 19.3 ± 16.4% greater, although statistically non-significant). Moreover, CCK-induced cortical LTP was more evident in CCK-deficient mice than C57 mice (*Chen et al., 2019a*).

These findings provide additional evidence for the rewiring and enhancement of adult thalamocortical connectivity through sensory experiences (*Montey and Quinlan, 2011*; *Blundon et al., 2017*; *Hogsden and Dringenberg, 2009*). Our results highlight the role of CCK in enhancing auditory thalamocortical plasticity and mitigating age-related deficits in frequency discrimination, underscoring its therapeutic potential in addressing auditory processing impairments in aging. However, PPI is a complex process that involves interactions between cortical and subcortical circuits. Future studies could explore how layer IV neurons contribute to these interactions and determine the specific roles of excitatory and inhibitory neurons in PPI performance. Such investigations will provide a more comprehensive understanding of the role of CCK in sensory gating and auditory processing.

## The potential role of CCK in modulating corticocortical plasticity via the thalamocortical pathway

Exogenous administration of CCK combined with tone exposure has been shown to enhance thalamocortical connectivity and improve frequency discrimination ability. However, it is important to consider that the observed changes in the ACx may not solely result from alterations at thalamocortical synapses. CCK released from entorhino-neocortical projections has been implicated in modulating

corticocortical plasticity and facilitating associative memory (*Chen et al., 2019a*; *Li et al., 2023*; *Sun et al., 2024*). Similarly, CCK released from thalamocortical projections may influence intracortical plasticity.

Following HFS of the MGB, the released CCK activates postsynaptic CCKBR, leading to an increase in intracellular calcium concentration through the activation of phospholipase C (*Lee et al., 1993*; *Detjen et al., 1997*; *Yule et al., 1999*). This calcium increase can facilitate the recruitment of AMPA receptors to the postsynaptic membrane (*Lynch et al., 1983*; *Malenka et al., 1988*; *Bredt and Nicoll, 2003*; *Wang et al., 1996*; *Berridge, 1998*). In the presence of CCK, repeated pre- and postsynaptic co-activation readily induces synaptic plasticity (*Li et al., 2014*; *Chen et al., 2019a*). The cortical neural response to MGB stimulation or acoustic stimulation involves both monosynaptic and poly-synaptic excitation and inhibition due to the complexity of the cortical network. Therefore, in addition to thalamocortical homosynaptic plasticity, CCK released from thalamocortical projections may modulate intracortical microcircuits in a heterosynaptic manner during activation. Considering the possibility of indirect activation, we specifically downregulated CCK expression to confirm that the released CCK originated from thalamocortical projection. While we focused on homosynaptic plasticity at thalamocortical synapses by recording only fEPSPs in layer IV of ACx, it is essential to further explore heterosynaptic effects of CCK released from thalamocortical synapses on intracortical circuits, particularly its role in modulating the excitatory-inhibitory balance. PV-interneurons, as key regulators of cortical inhibition, may contribute to the temporal fidelity of sensory processing, which is critical for auditory perception (*Nocon et al., 2023*; *Cai et al., 2018*). Additionally, CCK may facilitate cross-modal plasticity by modulating heterosynaptic plasticity in interconnected cortical areas. Future studies would provide valuable insights into the broader role of CCK in shaping sensory processing and cortical network dynamics.

## Conclusion

Thalamocortical plasticity in the adult brain has been found to be induced by different methods, reactivating plasticity in the sensory cortices beyond the critical period (*Blundon et al., 2017*; *Chung et al., 2017*; *Zhu et al., 2014*). Zhou et al. reported that exposing juvenile or adult rats to moderate noise levels reinstated the high susceptibility of the tonotopic map in the ACx (*Zhou et al., 2011*), while Blundon et al. demonstrated that restricting thalamic adenosine signaling could restore thalamocortical plasticity (*Blundon et al., 2017*). In parallel, our current study proposes CCK administration as a promising approach to induce thalamocortical plasticity in the adult brain, providing a foundation for exploring CCK-based interventions in age-related sensory and cognitive deficits. CCK-mediated plasticity holds promise as a treatment strategy to enhance the effectiveness of late cochlear implants in prelingually deaf children (*Niparko et al., 2010*) and may serve as a basis for future non-invasive therapies targeting age-related hearing loss. Thalamocortical plasticity is not only involved in refining sensory cortices but also plays a crucial role in learning processes (*Biane et al., 2016*). Therefore, our findings may have broader implications across various sensory modalities, including perception, learning, and memory formation.

## Methods

### Animals

In the present study, male mice and rats were used to investigate thalamocortical LTP. Experiments were conducted using C57BL/6 wild-type (C57) mice of different ages (neonatal: P14 and P20; young adult: 8 weeks, P54-P56; adult: 3–4 months; aged: 17–19 months), CCK-ires-Cre (Jax#019021, $Cck^{Cre}$), $Cck^{CreERT2}$ (Jax#012710, CCK-deficient; in our cohort, RNAscope detected no *Cck* mRNA in the MGB and ACx (*Figure 3A and B*), therefore we refer to these mice operationally as 'CCK-deficient' throughout.) and Sprague-Dawley rats. Animals were housed at 20–24°C with 40–60% humidity under a 12-hr-light/12-hr-dark cycle (lights off from 8:00 am to 8:00 pm) with free access to food and water. All experimental procedures were approved by the Animal Subjects Ethics Sub-Committees of the City University of Hong Kong.

### Chemicals and antibodies

For in vivo electrophysiological experiments, CCK-4 (Cat. No. T6515) was purchased from Sigma. CCK-8S (Cat.No. 1166), a selective CCKBR antagonist, L-365, 260 (Cat. No. 2767), DMSO (Cat. No.

3176) were purchased from Tocris Bioscience (Hong Kong, SAR). Artificial cerebrospinal fluid (ACSF, item# 59–7316) was purchased from Harvard Apparatus (U.S.) and used as the solvent for the antagonist above. For immunohistochemical experiments, mouse anti-PSD95 (Invitrogen, #MA1-045, 1:500), goat anti-CCKBR (Invitrogen, #PA5-18384, 1:1000). Secondary antibodies included donkey anti-mouse IgG(H+L) Alexa Fluor 594 (Invitrogen, #A-11058, 1:500), donkey anti-goat IgG(H+L) Alexa Fluor 647 (Invitrogen, #A-21447, 1:500).

## Surgery for acute electrophysiological experiments

Mice or rats were anesthetized with urethane sodium (1.8 g /kg IP; Sigma, U.S.). Anesthesia was maintained throughout surgery. Atropine sulfate (0.05 mg/kg SC; Sigma, U.S.) was administered 15 min before the induction of anesthesia to inhibit tracheal secretion. The animal was mounted in a stereotaxic frame (Narishige, Japan) and a midline incision was made in the scalp after a liberal application of a local anesthetic (Xylocaine, 2%). A craniotomy was performed at the temporal bone, as the auditory cortex is located on the lateral surface of the brain (coordinates: 1.5–3.0 mm below the temporal ridge and 2 mm to 4 mm posterior to bregma for mice; 2.5–6.5 mm below the temporal ridge and 3.0–5.0 mm posterior to bregma for rats) to access the auditory cortex, and a hole was drilled in the skull according to the coordinates of the ventral division of the MGB (MGv, AP: –3.2 mm, ML: 2.1 mm, DV: 3.0 mm) for experiments conducted on mice. High-magnification photomicrographs were acquired to confirm that electrode tips were positioned within the MGv (*Figure 1A*). The dura mater was minimally opened, followed by the silicone oil application to the surface of the brain to prevent drying. The animal's body temperature was maintained at 37–38°C with a feedback-controlled heating pad (RWD, China). After the recording, animals were sacrificed, and the brains were harvested for histological confirmation or further processing.

## Virus and retrograde tracer injection

Mice were anesthetized with pentobarbital (50 mg/kg i.p., France) and kept under anesthetic status by supplying one-third of the initial dosage once per hour. For the virus injection into MGB, two small holes were drilled bilaterally in the skull according to the coordinates of the ventral division of the MGB (MGv subdivision, AP: –3.2 mm, ML: 2.1 mm, DV: 3.0 mm). In the optogenetic electrophysiological experiment shown in *Figure 2A*, AAV9-EFIa-DIO-ChETA-EYFP (300 nL, 6.00E12 gc/mL, Molecular Tools Platform, Canada) was injected into the MGB of $Cck^{Cre}$ mice at a rate of 30 nL/min (Nanoliter Injector, World Precision Instruments). In the shRNA electrophysiological experiment shown in *Figure 2C*, rAAV-hSyn-EGFP-5'miR-30a-shRNA($Cck$)–3'-miR30a-WPREs (400 nL, 5.63E12 gc/mL, BrainVTA, China) or rAAV-hSyn-EGFP-5'miR-30a-shRNA(Scramble)–3'-miR30a-WPREs (400 nL, 6.08E12 gc/mL, BrainVTA, China) was injected into the MGB of C57 mice. In the fiber photometry experiment shown in *Figure 2E*, AAV9-Syn-FLEX-ChrimsonR-tdTomato (300 nL, 4.00E12 gc/mL, Addgene, U.S.) /AAV9-Syn-ChrimsonR-tdTomato (300 nL, 4.15E12 gc/mL, Addgene, U.S.) was injected into the MGB of $Cck^{Cre}$/CCK-deficient mice. In the shRNA fiber photometry experiment shown in *Figure 2H*, rAAV-hSyn-mCherry-5'miR-30a-shRNA($Cck$)–3'-miR30a-WPREs (400 nL, 5.64E12 gc/mL, BrainVTA, China) or rAAV-hSyn-mCherry-5'miR-30a-shRNA(Scramble)–3'-miR30a-WPREs (400 nL, 5.55E12 gc/mL, BrainVTA, China) was injected into the MGB of C57 mice. In the optogenetic electrophysiological experiment shown in *Figure 4D* and *Figure 5—figure supplement 1B*, AAV9-Syn-ChrimsonR-tdTomato (300 nL, 6.50E12 gc/mL, Addgene, U.S.) was injected into the MGB of C57 mice.

For the ACx injection, two small holes were drilled bilaterally in the skull (AP: –2.8 mm posterior to bregma, ML: –4.2 mm lateral to the midline, DV: 0.9 mm below the pia) to approach ACx to inject AAV9-hSyn-CCKsensor (500 nL, 4.81E12 gc/mL, BrainVTA, China; *Figure 2E and H*, *Figure 3E*, and *Figure 5—figure supplement 1F*). After injection, animals were sutured and returned to their home cages for recovery. Antiseptic and analgesic balm was applied on the surface of the wound during the first 3 days after the surgery.

To minimize viral spread, several strategies were implemented to ensure targeted delivery. These included the use of fine-tipped injection needles, tissue stabilization for 7 min after needle insertion, delivery of small volumes at a slow rate (30 nL/min) to prevent backflow, aspiration of 5 nL of the solution post-injection, and gradual retraction of the needle by raising it 100 μm and waiting an additional 5 min before full retraction. After the electrophysiological experiments, the accuracy of viral

expression was systematically verified by performing post-hoc histological analyses to ensure that the expression was primarily localized within the intended regions.

## Auditory, electrical, and laser stimuli

Auditory stimuli, including pure tones and noise bursts, were generated by Tucker-Davis Technologies (TDT, U.S.) workstation and delivered through an electrostatic speaker (ED1, TDT). The speaker was placed 20 cm away from the awake animals or directly to the ear contralateral to the implanted electrodes via a hollow ear bar for the anesthetized animals. The sound pressure level of the speaker was calibrated with a condenser microphone (B&K, Denmark).

ES was generated using an ISO-Flex isolator (A.M.P.I., Israel) controlled by a multifunction processor (RX6, TDT). The electrical current pulses for the baseline test were 0.5 ms, 10–100 μA, and were delivered every 10 s. The HFS contained four trains of 10 bursts at 5 Hz with an interval of 10 s between two trains, and each burst consisted of 5 pulses at 100 Hz.

The laser stimulation was produced by a laser generator (5–20 mW, Wavelength, 473 nm, 620 nm; CNI laser, China) controlled by an RX6 system and delivered to the brain via an optic fiber (Thorlabs, U.S.) connected to the generator (*Sun et al., 2024*). The output power of the fiber was measured and calibrated by an optical power meter (Item# PM120A, Thorlabs, U.S.) before the insertion into the brain. The laser pulse width was 3ms, and the interval for baseline testing was 10 s. For the HFLS, the laser stimulation was comprised of four trains of 10 bursts at 5 Hz with an interval of 10 s between the trains, and each burst consisted of five pulses at 80 Hz.

## In vivo acute electrophysiological recording

In HFS-induced thalamocortical LTP experiments, two customized microelectrode arrays with four tungsten unipolar electrodes each, impedance: 0.5–1.0 MΩ (recording: CAT.# UEWSFGSECNND, FHC, U.S.), and 200–500 kΩ (stimulating: CAT.# UEWSDGSEBNND, FHC, U.S.), were used for the auditory cortical neuronal activity recording and MGB ES, respectively. The electrode arrays were advanced to the brain by two micro-manipulators separately. The recording electrodes were lowered into layer IV of ACx, while the stimulation electrodes were lowered into MGB (MGv subdivision). The final stimulating and recording positions were determined by maximizing the cortical fEPSP amplitude triggered by the ES in the MGB. The accuracy of electrode placement was verified through post-hoc histological examination and electrophysiological responses.

The fEPSPs were elicited by 0.5 ms electrical current pulses. They were amplified (×1000), filtered (1 Hz –5 kHz), and recorded at a 25 kHz sampling rate. The data were stored in a PC using OpenEx software (TDT). Before the recording, an input-output function was measured. A stimulation current, which elicited a fEPSP amplitude 40% of maximum, was chosen for baseline and post-HFS recording. The fEPSPs were collected for 16 min before and 1 hr after HFS. For HFS, each burst includes five 0.5 ms pulses at 100 Hz, and each block consists of 10 bursts at 5 Hz, for a total of four blocks with an inter-block interval of 10 s. The current of the pulses that induced 75% of the maximal response was selected from the input-output relationship. The slopes of the evoked fEPSPs were calculated and normalized using a customized MATLAB script (source code file is included in the submission), and the group data were plotted as mean ± SEM. In the experiment shown in *Figure 1*, 50 noise bursts (Intensity, 70 dB; Duration, 100 ms; Inter-stimulus-interval, 10 s) were presented before and after LTP induction session, and multiunit responses to noise bursts of the ACx were recorded. A threshold of 3 SDs above baseline was set to identify spikes online.

## Optogenetic experiments

In the optogenetic experiments, glass pipette electrodes were placed targeting layer IV (350–500 μm) of the ACx of mice 4–6 weeks after virus injection. The optic fiber was then inserted into either the ACx next to the electrodes, or the MGB (*Sun et al., 2024*). The fEPSPs evoked by laser stimulation were recorded and analyzed in this experiment. The procedure for the HF laser-induced LTP was similar to the HFS-induced LTP experiment, except the HF burst containing laser pulses in 80 Hz rather than 100 Hz.

## Drug infusion experiments

A glass pipette was placed to ACx adjacent to the recording site for drug application. The tips of glass pipettes were covered by 0.1 μL silicone oil to avoid leaking.

In the antagonist infusion experiments, L-365,260 (250 nM in 5% DMSO, 0.5 µL, Tocris) was injected into the ACx using a micro-injector 5 min before HFS. ACSF (5% DMSO) was injected as a control. In the CCK infusion experiments, CCK-4 (10 µM, 0.5 µL, Sigma) was injected using a micro-injector over a 5-min period. ACSF was injected as a control. In the rat experiments, CCK-8S (1 µM, 1 µL, Tocris) or ACSF was infused into the ACx.

## Auditory tuning curve test on rats

In the tuning curve test, tones spanning 6 octaves (0.75–48 kHz, 0.2–0.3octave spacing) and 60 dB (10–70 dB, 5–10 dB spacing) of 100ms duration were presented every 500ms in a pseudo-random sequence before and after the infusion and LFS to measure the receptive field of cortical neurons. The tuning curve was determined by plotting the lowest intensity at which the neuron responded to different tones. The characteristic frequency (CF) is defined as the frequency corresponding to the lowest point on this curve. The effective frequency (EF) was determined to elicit a clear sound response while maintaining a sufficient distance from the CF to allow measurable increases in response intensity. Specifically, EF was selected based on the starting point of the tuning peak, which corresponds to the onset of its fastest rising phase. From this point, EF was determined by moving 0.2 or 0.4 octaves toward the CF. For the infusion and LFS protocol, CCK-8S (1 µM, 1 µL) or ACSF was infused locally near the recording site, and EF tones were presented once per 2 s for 200 trials, 5 min after the infusion. The 30ms before the onset was considered as the baseline, and the 30ms after the onset of each tone was considered as onset response. The potential tone response bins, of which the firing rates were larger than the averaged firing rate of all the baseline bins plus 3 SDs of the firing rates of these baseline bins (mean + 3 SD), were considered to have tone response. To quantify the changes in the tuning curve, we used the frequency responding threshold as the indicator of the change. The data were plotted as mean ± SEM. After the recording, animals were sacrificed, and the brains were harvested for histological confirmation or immunostaining.

## Fiber photometry

This GPCR activation-based CCK sensor, GRAB-CCK, was developed by inserting a circular-permutated green fluorescent protein (cpEGFP) into the intracellular domain of CCKBR (*Wang et al., 2023*). When the endogenous or exogenous ligand (CCK) binds to the CCKBR, a conformational change in cpEGFP will happen as well as an increase in fluorescence intensity, and the CCK activity could be visualized in vivo. Six weeks after CCK-sensor virus injection, a craniotomy was performed to access the auditory cortex at the temporal bone (1.5–3.0 mm below the temporal ridge and 2.0–4.0 mm posterior to bregma), and the dura mater was opened. An optic fiber (400 µm diameter, 0.22 NA, Thorlabs, Newton, NJ) was lowered into the auditory cortex (200–300 µm from the brain surface) to record the signal of the CCK sensor. Before signal recording, the optic fiber was lowered to the brain surface at different sites to confirm the best site for the recording, where we could capture the strongest fluorescence signal of CCK sensor. This optic fiber cannula was attached to a single fluorescent MiniCube (Doric Lenses, Quebec, QC, Canada) with built-in dichroic mirrors. LED light sources were connected through a fiber patch cord. The excitation light at 470 and 405 nm was released by two fiber-coupled LEDs (M470F3 and M405FP1, Thorlabs) and was sinusoidally modulated at 210 and 330 Hz, respectively. The 470 nm channel is the GRAB-CCK channel, and the 405 nm channel is employed as the isosbestic control channel. An LED driver (LEDD1B, Thorlabs) coupled to the RZ5D processor (TDT, Alachua, FL) managed the excitation light's intensity through the software Synapse. The emission fluorescence was captured and transmitted by a bandpass filter in the MiniCube. To avoid photobleaching, the excitation light intensity at the tip of the patch cord's tip was adjusted to less than 30 µW. The fluorescent signal was then detected, amplified, and transformed into an analog signal by the photoreceiver (Doric Lenses). The analog signal was then digitalized by the RZ5D processor and subjected to a 1 kHz low-pass analysis using Synapse software.

The fluorescent signal was monitored for 25 s before and 60 s after the HFLS (5–10 mW, 620 nm) or HFS application.

For the experiments in *Figure 2E–G*, CCK-deficient mice were used to validate the specificity of the CCK sensor by comparing fluorescence intensity changes post-HFS with those observed in *Cck*Cre mice. This allowed us to confirm that the observed fluorescence changes were specifically due to CCK release and not other neurotransmitters.

For the experiments in *Figure 5—figure supplement 1G*, we measured the CCK sensor activities before and after the CCK-4 i.p. application in different types of mice. CCK-4 (1 mg/kg, in 2% DMSO, 8% ethylene glycol, 1% Tween-80, 89% saline) or saline (0.15–0.20 mL, in 2% DMSO, 8% ethylene glycol, 1% Tween-80) was i.p. injected. The fluorescent signal was recorded 120 s before and 200 s after the drug administration.

## Fiber photometry analysis

Analysis of the signal was done by the custom-written MATLAB (Mathworks) codes. We first extracted the signal of the 470 nm and 405 nm channels corresponding to the defined periods before and after each stimulus or drug injection. A fitted 405 nm signal was created by regressing the 405 nm channel onto a linear fit of its respective 470 channel (MATLAB *polyfit* function). The fluorescence change (ΔF/F) was then calculated with the formula (470 nm signal − fitted 405 nm signal)/ fitted 405 nm signal.

## Immunohistochemistry

Mice were anesthetized by an overdose of pentobarbital sodium and transcardially perfused with 30 mL cold phosphate-buffered saline (PBS) and 30 mL 4% (w/v) paraformaldehyde (PFA) in PBS. Brain tissue was removed and post-fixed overnight in 4% PFA in PBS at 4°C, and treated with 30% (w/v) sucrose in PBS at 4°C for 2 days. Coronal sections of 50 μm thickness were cut on a cryostat (Epredia CryoStar HM525 NX Cryostat) using OCT and preserved with antifreeze buffer (20% (v/v) glycerol and 30% (v/v) ethylene glycol diluted in PBS) at −20°C. Brain sections were washed three times with PBS before being incubated in blocking buffer containing 5% (v/v) normal goat serum (or 5% (w/v) bovine serum albumin if the host of primary antibody is goat) with 0.3% (v/v) Triton X-100 in PBS (PBST) for 2 hr at room temperature. and then incubated with primary antibodies in blocking buffer for 36 hr at 4°C. The primary antibodies used were: mouse anti-PSD95 (Invitrogen, #MA1-045, 1:500), goat anti-CCKBR (Invitrogen, #PA5-18384, 1:1000). After three washes in PBS, sections were incubated with fluorescently conjugated secondary antibodies: donkey anti-mouse IgG(H+L) Alexa Fluor 594 (Invitrogen, #A-11058, 1:500), donkey anti-goat IgG(H+L) Alexa Fluor 647 (Invitrogen, #A-21447, 1:500) in PBST at room temperature for 2.5 hr. Next, sections were incubated in DAPI (Chem Cruz, #SC3598) for 5 min, washed several times, and mounted with 70% (v/v) glycerol in PBS on slides. Images of immunostained sections were acquired on Nikon confocal microscope and processed with NIS-Element (Nikon) and ImageJ (NIH). For the colocalization analysis in *Figure 2A*, we first merged channels of virus AAV9-EFIa-DIO-ChETA-EYFP (green) and PSD95 (red), and the co-labeled yellow areas are considered as CCK-positive terminals. The areas were filtered by Color Threshold and then merged with CCKBR channel (magenta).

## RNAscope in situ hybridization

Coronal brain sections (20 μm) were prepared with the same methods as tissue used for immunohistochemistry (see above). Using mouse-specific CCK probe (ACDbio, #402271) and RNAscope reagent kit-RED (ACDbio, #322350), we performed chromogenic in situ hybridization according to the manufacturer's instructions for fixed frozen sections. Briefly, washed sections were baked for 30 min at 60°C, then fixed in 4% PFA in PBS for 15 min at 4°C and dehydrated in successive 5 min baths of ethanol (50, 75, 100%). After drying, three steps of pretreatment were performed, including a 10-min hydrogen peroxide treatment, a 15-min target retrieval step in boiling solution, and a 16-min protease digestion step at 40°C. After creating a hydrophobic barrier around the perimeter of each section, hybridization with CCK probe was performed for 2 hr at 40°C, followed by six steps of amplification. Two washes of 2 min were applied after the hybridization step and each amplification step. Fast Red was then used as a chromogen for the exposure step for 10 min at room temperature. After washing in water, slides were counterstained with freshly diluted hematoxylin, washed again and dried, then mounted with Vectamount (ACObio, #321584) on SuperFrost Plus slides (Thermo Fisher Scientific, #12-550-15).

It is understood that higher sensitivity can be achieved using the in situ hybridization fluorescent assays. However, the objective in assessing *Cck* mRNA expression level was to compare the differences across several developmental stages. Therefore, identical parameters for imaging and image analysis were applied to ensure accurate comparisons. Brightfield images were acquired using a Nikon Eclipse Ni-E upright microscope. Semi-quantitative analysis of the images was performed employing

Fiji ImageJ software. ROI was delineated separately for each slice, focusing on the MGv region. The background was subtracted using the Color Deconvolution with the same user-defined values for all the calculated images. Images from the signal channel were first converted to the eight-bit grayscale before applying a threshold as a criterion for binarization, where signals equal to or above the threshold were considered positive. It should be noted that the positive cells we defined here are based on this criterion rather than an absolute definition, and the specific threshold value was appropriately chosen to identify the most of CCK-positive neurons in the adult group. Overlapping cells were then separated using the Watershed segmentation function prior to running the Analyze Particles tool for cell counting. The signal intensity of each single cell was then measured on the eight-bit grayscale images, according to the location and area of the positive cells confirmed in the binary images. The *Cck* mRNA intensity of a single cell was normalized to the average *Cck* expression of all the MGv neurons from 8-week mice. For CCK-deficient mice, the normalized CCK mRNA intensity was essentially zero (*Figure 3B*).

## Cannula implantation

C57 mice were anesthetized, and the scalp was opened, as mentioned above. Two small holes were prepared bilaterally based on the location of the primary ACx (AP: –3.0 mm, ML: 4 mm, DV: 1.2 mm). Drug injection cannulas with metallic caps and dummies (Length: 6 mm, Diameter: 0.6 mm., RWD, China) were inserted into the primary ACx bilaterally and then fixed with C&B-MetaBond adhesive luting cement (Parkell, U.S.) and dental cement (Mega Press, Germany). After the implantation, the animals were returned to their home cages for recovery. Antiseptic and analgesic balm was applied on the surface of the wound on the first 3 days after the surgery.

## Frequency discrimination test

### For young adult mice group shown in *Figure 5A-D*

Prepulse inhibition of the acoustic startle response test was adopted to examine the frequency discrimination ability of mice in CCK-8S injection group and ACSF control group.

In the test, three customized soundproof chambers equipped with vibration sensors on the bottoms for the startle reflex detection, MF-1 multi-field magnetic speakers for tone presentation, and high-power tweeter speakers for startle white noise presentation. Three days after the cannula implantation surgery, mice were placed into the self-designed plastic tubes individually with open slots on both sides and front for habituation for 5–10 min every day for 3 days. The mice were then randomly separated into two groups: one CCK-8S infusion group and one ACSF infusion control group. CCK-8S (10 nM, 1 µL) or ACSF (1 µL) was infused by microinjector at the speed of 0.2 µL/min bilaterally into the auditory cortices. Five minutes after the injection, mice were placed in the tubes and exposed to pure-tone stimulation for 0.5 hr in the sound-proof chamber. For the tone exposure protocol, a pure tone of 9.8 kHz or 16.4 kHz (Intensity, 70 dB; duration, 100ms) was used. The tone exposure consisted of 900 trains with an inter-tone interval of 2 s. Each train contained five tones presented at 5 Hz. Mice were restrained into the tubes again for the startle test at 24 hr after the exposure. The tubes were stabilized on the vibration sensors in the soundproof chamber during the experiment. The startle reflexes were detected by the sensors and then amplified and recorded by TDT. The whole experiment was divided into four blocks; a background tone (9.8 kHz or 16.4 kHz) was continuously presented at 70 dB throughout the experiment. Block 1 was a 5-min acclimation period in which only the background tone was presented. Block 2 contained nine startle-only trials in which a white noise burst of 120 dB intensity and 20ms duration was presented. Block 3 consisted of prepulse inhibition trials in a pseudorandom order. Each prepulse trial consisted of a 80ms prepulse at 70 dB (pure-tone frequency was 0%, 2%, 4%, 8%, 16%, or 32% lower than the background tone, Δf), followed by a 20 ms white-noise startle pulse at 120 dB, and then return to the background tone after the startle. Every trial in Block 3 was presented 15 times. Block 4 was identical to Block 2, and it was used to detect any habituation within the experiment. The inter-trial interval was randomized between 10 and 20 s. The startle reflex was measured as the peak-to-peak amplitude of the raw waveform detected by the sensor. Prepulse inhibition percentage was calculated from bBlock 3 data as follows: [1 – (response amplitude in prepulse trial/ response amplitude in Δf=0% prepulse trial)]×100, and the data was plotted as mean ± SEM. Startle reflex amplitude in Block 2 and Block 4 were compared with each other as an internal

control for startle attenuation over the whole experiment. Mice with statistically different performance in Block 2 and Block 4 were removed from the analysis.

### For the aged mice group shown in *Figure 5E and F*

Thirty aged C57 mice of 17–19 months were divided into two groups: Experiment group (CCK-4) and their littermate control group (saline).

The PPI protocol was the same except i.p. CCK-4 (1 mg/kg, in 2% DMSO, 8% ethylene glycol, 1% Tween-80) or saline (2% DMSO, 8% ethylene glycol, 1% Tween-80) injection instead of directly delivering the drug into ACx by implanted cannula. 9.8 kHz was used as the background tone and the exposed tone. Tone exposure session consisted of 300 trains with an inter-tone interval of 2 s. Each train included five tones (9.8 kHz, 100ms) presented at 5 Hz. Since the half-life of CCK-4 is short, CCK-4 injection accompanied by pure tone exposure was performed three times in D3.

## Data analysis

All data are presented as mean ± SEM. Error bars and shaded areas represent SEM. Here, n represents the number of stimulation-recording sites and N represents the number of animals in each experiment. At each time point, fEPSPs were averaged across 12 consecutive trials (2 min). All statistical analyses (one-way RM ANOVA or two-way RM ANOVA) were done in SPSS (IBM, USA). Pairwise comparisons were adjusted by Bonferroni correction. The Tukey test was used in the behavioral study for the post-hoc test. p-Values <0.05 were considered statistically significant.

## Acknowledgements

We thank Prof. Tomas Hökfelt (Karolinska Institutet) and Prof. Bin Hu (Calgary) for critical reviews. We also thank Prof. Kuan-Hong Wang (Rochester), Prof. Xiaomin Zhou (East China Normal University) and Prof. Jan Schnupp (CityU) for their comments on Jingu Feng's Ph.D. thesis (City University of Hong Kong, 2018), which contributed to the foundation of the current manuscript. We thank Prof. Yulong Li (Peking University) for providing the CCK-sensor virus. This work was supported by Hong Kong Research Grants Council and Health and Medical Research Fund, Innovation and Technology Fund (C1014-15G, C1002-24W, C5053-22G, SRFS2324-1S02, MRP/101/17 X, MPF/053/18 X, 08194106, 03141196, 01121906, 09203656, 11101215 M, 11166316 M, 11102417 M, 11101818 M, 11103922, 11104923, 11104524). We also thank the following charitable foundations for their generous support: Wong Chun Hong, Charlie Lee Charitable Foundation, Fong Shu Fook Tong Foundation, and Croucher Foundation.

## Additional information

### Funding

| Funder | Grant reference number | Author |
| --- | --- | --- |
| University Grants Committee | C1014-15G | Jufang He |
| Innovation and Technology Commission | MRP/101/17X | Jufang He |
| Innovation and Technology Commission | MPF/053/18X | Jufang He |
| Health Bureau | 08194106 | Xiao Li |
| Health Bureau | 03141196 | Jufang He |
| Health Bureau | 01121906 | Jufang He |
| Research Grants Council, University Grants Committee | 11101215M | Jufang He |

| Funder | Grant reference number | Author |
|---|---|---|
| Research Grants Council, University Grants Committee | 11166316M | Jufang He |
| Research Grants Council, University Grants Committee | 11102417M | Jufang He |
| Research Grants Council, University Grants Committee | 11101818M | Jufang He |
| Research Grants Council, University Grants Committee | 11103922 | Jufang He |
| Research Grants Council, University Grants Committee | 11104923 | Jufang He |
| Research Grants Council, University Grants Committee | 11104524 | Jufang He |
| Research Grants Council, University Grants Committee | C1002-24W | Jufang He |
| Research Grants Council, University Grants Committee | C5053-22G | Jufang He |
| Research Grants Council, University Grants Committee | SRFS2324-1S02 | Jufang He |
| Health Bureau | 09203656 | Jufang He |

The funders had no role in study design, data collection and interpretation, or the decision to submit the work for publication.

## Author contributions

Xiao Li, Conceptualization, Data curation, Software, Formal analysis, Supervision, Funding acquisition, Validation, Investigation, Visualization, Methodology, Writing – original draft, Project administration, Writing – review and editing; Jingyu Feng, Conceptualization, Data curation, Formal analysis, Validation, Investigation, Visualization, Methodology, Writing – original draft; Xiaohan Hu, Data curation, Formal analysis, Validation, Investigation, Visualization, Writing – review and editing; Peipei Zhou, Data curation, Formal analysis, Investigation, Writing – review and editing; Tao Chen, Xuejiao Zheng, Formal analysis, Validation, Investigation; Peter Jendrichovsky, Resources, Data curation, Formal analysis, Validation, Methodology; Xue Wang, Formal analysis, Validation, Investigation, Writing – review and editing; Mengying Chen, Hao Li, Validation, Investigation; Xi Chen, Conceptualization, Software, Funding acquisition, Validation, Methodology; Dingxuan Zeng, Mengfan Zhang, Investigation; Zhoujian Xiao, Validation; Ling He, Visualization, Writing – review and editing; Stephen Temitayo Bello, Writing – review and editing; Jufang He, Conceptualization, Resources, Supervision, Funding acquisition, Methodology, Project administration, Writing – review and editing

## Author ORCIDs

Xiao Li https://orcid.org/0000-0003-4067-6980
Xi Chen https://orcid.org/0000-0002-2144-6584
Mengfan Zhang https://orcid.org/0000-0002-5320-5742
Ling He https://orcid.org/0000-0001-8723-7702
Jufang He https://orcid.org/0000-0002-4288-5957

## Ethics

All experimental procedures were approved by the Animal Subjects Ethics Sub-Committees of the City University of Hong Kong.

Reviewer #1 (Public review): https://doi.org/10.7554/eLife.101513.4.sa1
Reviewer #2 (Public review): https://doi.org/10.7554/eLife.101513.4.sa2
Reviewer #3 (Public review): https://doi.org/10.7554/eLife.101513.4.sa3
Author response https://doi.org/10.7554/eLife.101513.4.sa4

## Additional files

### Supplementary files
Source data 1. Source data for all figures.

Source code 1. Matlab code for data analysis.

MDAR checklist

### Data availability
All data generated or analyzed during this study are included in the manuscript and supporting files; source data files have been provided for all figures. Further information and requests for resources and reagents should be directed to and will be fulfilled by the Lead Contact, Jufang He (jufanghe@ cityu.edu.hk).

The following dataset was generated:

| Author(s) | Year | Dataset title | Dataset URL | Database and Identifier |
|---|---|---|---|---|
| Li X, Feng J, Hu X, Zhou P, He J | 2025 | Cholecystokinin modulates age-dependent thalamocortical Neuroplasticity | https://doi.org/ 10.5061/dryad. k0p2ngfmt | Dryad Digital Repository, 10.5061/dryad.k0p2ngfmt |

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
