## [Editor Report · eLife Assessment]

This is an **important** study demonstrating that cholecystokinin is a key modulator of auditory thalamocortical plasticity during development and in young adult but not aged mice, though cortical application of this neuropeptide in older animals appears to go some way to restoring this age-dependent loss in plasticity. A strength of this work is the use of multiple experimental approaches, which together provide **convincing** support for the proposed involvement of cholecystokinin. This work is likely to be influential in opening up a new avenue of investigation into the roles of neuropeptides in sensory plasticity.

---

## [Referee Report · Reviewer #1 (Public review)]

This report addresses a compelling topic. The authors demonstrate that tetanic stimulation of the auditory thalamus induces cortical long-term potentiation (LTP), which can be elicited by either electrical or optical stimulation of the thalamus or by noise bursts. They further show that thalamocortical LTP is abolished when thalamic CCK is knocked down or when cortical CCK receptors are blocked. Notably, in 18-month-old mice, thalamocortical LTP was largely absent but could be restored by cortical application of CCK. The authors conclude that CCK is a critical contributor to thalamocortical plasticity and may enhance this form of plasticity in aged subjects.

The findings presented in this report are valuable and advance our understanding of thalamocortical plasticity.

---

## [Referee Report · Reviewer #2 (Public review)]

Summary:

This work used multiple approaches to show that CCK is critical for long-term potentiation (LTP) in the auditory thalamocortical pathway. They also showed that the CCK mediation of LTP is age-dependent and supports frequency discrimination. This work is important because is opens up a new avenue of investigation of the roles of neuropeptides in sensory plasticity.

Strengths:

The main strength is the multiple approaches used to comprehensively examine the role of CCK in auditory thalamocortical LTP. Thus, the authors do provide a compelling set of data that CCK mediates thalamocortical LTP in an age-dependent manner.

Weaknesses:

The behavioral assessment is relatively limited, but may be fleshed out in future work.

---

## [Referee Report · Reviewer #3 (Public review)]

Summary:

Cholecystokinin (CCK) is highly expressed in auditory thalamocortical (MGB) neurons and CCK has been found to shape cortical plasticity dynamics. In order to understand how CCK shapes synaptic plasticity in the auditory thalamocortical pathway, they assessed the role of CCK signaling across multiple mechanisms of LTP induction with the auditory thalamocortical (MGB - layer IV Auditory Cortex) circuit in mice. In these physiology experiments that leverage multiple mechanisms of LTP induction and a rigorous manipulation of CCK and CCK-dependent signaling, they establish an essential role of auditory thalamocortical LTP on the co-release of CCK from auditory thalamic neurons. By carefully assessing the development of this plasticity over time and CCK expression, they go on to identify a window of time that CCK is produced throughout early and middle adulthood in auditory thalamocortical neurons to establish a window for plasticity from 3 weeks to 1.5 years in mice, with limited LTP occurring outside of this window. The authors go on to show that CCK signaling and its effect on LTP in the auditory cortex is also capable of modifying frequency discrimination accuracy in an auditory PPI task. In evaluating the impact of CCK on modulating PPI task performance, it also seems that in mice <1.5 years old CCK-dependent effects on cortical plasticity is almost saturated. While exogenous CCK can modestly improve discrimination of only very similar tones, exogenous focal delivery of CCK in older mice can significantly improve learning in a PPI task to bring their discrimination ability in line with those from young adult mice.

Strengths:

(1) The clarity of the results along with the rigor multi-angled approach provide significant support for the claim that CCK is essential for auditory thalamocortical synaptic LTP. This approach uses a combination of electrical, acoustic, and optogenetic pathway stimulation alongside conditional expression approaches, germline knockout, viral RNA downregulation and pharmacological blockade. Through the combination of these experimental configures the authors demonstrate that high-frequency stimulation-induced LTP is reliant on co-release of CCK from glutamatergic MGB terminals projecting to the auditory cortex.

(2) The careful analysis of the CCK, CCKB receptor, and LTP expression is also a strength that puts the finding into the context of mechanistic causes and potential therapies for age-dependent sensory/auditory processing changes. Similarly, not only do these data identify a fundamental biological mechanism, but they also provide support for the idea that exogenous asynchronous stimulation of the CCKBR is capable of restoring an age-dependent loss in plasticity.

(3) Although experiments to simultaneously relate LTP and behavioral change or identify a causal relationship between LTP and frequency discrimination are not made, there is convincing evidence that CCK signaling in the auditory cortex (known to determine synaptic LTP) is important for auditory processing/frequency discrimination. These experiments are key for establishing the relevance of this mechanism.

Weaknesses:

The following are weaknesses or limitations of the study that may also fall outside of the scope of this work, but which could be addressed in the future.

(1) Given the magnitude of the evoked responses, one expects that pyramidal neurons in layer IV are primarily those that undergo CCK-dependent plasticity, but the degree to which PV-interneurons and pyramidal neurons participate in this process differently is unclear.

(2) While these data support an important role for CCK in synaptic LTP in the auditory thalamocortical pathway, perhaps temporal processing of acoustic stimuli is as or more important than frequency discrimination. Given the enhanced responsivity of the system, it is unclear whether this mechanism would improve or reduce the fidelity of temporal processing in this circuit. Understanding this dynamic may also require consideration of cell type as raised in weakness #1.

(3) In Figure 1, an example of increased spontaneous and evoked firing activity of single neurons after HFS is provided. Yet it is surprising that the group data are analyzed only for the fEPSP. It seems that single neuron data would also be useful at this point to provide insight into how CCK and HFS affect temporal processing and spontaneous activity/excitability.

---

## [Author Response]

The following is the authors’ response to the previous reviews

**Reviewer #1 (Public review):**
This report addresses a compelling topic. However, I have significant concerns, which necessitate a reassessment of the report's overall value.Anatomical Specificity and Stimulation Site:While the authors clarify that the ventral MGB (MGv) was the intended stimulation target, the electrode track (Fig. 1A) and viral spread (Fig. 2E) suggest possible involvement of the dorsal MGB (MGd) and broader area. Given that MGv-AI and MGd-AC pathways have distinct-and sometimes opposing-effects on plasticity, the reported LTP values (with unusually small standard deviations) raise concerns about the specificity of the findings. Additional anatomical verification would help resolve this issue.

We thank the reviewer for highlighting the importance of anatomical specificity in MGv targeting. In the revised manuscript, we have taken several steps to address these issues:

(1) Higher-magnification histology has been added to Figure 1A, clearly identifying the electrode tip localized within the MGv.

(2) Figure 2E has been replaced with a new image showing viral expression largely confined to MGB, with minimal spread to surrounding structures.

(3) In the Discussion, we explicitly acknowledge that although targeting was guided by stereotaxic coordinates and histological confirmation, some viral spread throughout the MGB occurred. We also discuss the possibility that both MGv-A1 and MGd-AC pathways may contribute to the recorded responses, which could influence the observed plasticity, as previously suggested by the reviewer.

These additions and acknowledgments are now incorporated to ensure the reader can interpret the data with full consideration of anatomical targeting limitations.

Results section:

“Higher-magnification histology confirmed accurate MGv targeting (Figure 1A, lower-middle panel)’”

Discussion section:

“Although our experiment targeting the MGv was guided by stereotaxic coordinates and verified post hoc, we acknowledge potential contributions from non-lemniscal medial geniculate nucleus dorsal (MGd) projections. Anatomical and physiological evidence indicates that MGv-AC projections provide rapid, frequency‑specific, tonotopically organized excitation, whereas MGd pathways target higher‑order auditory cortex with broader tuning, less precise tonotopy, longer response latencies, and greater context‑dependence, features that can differentially shape cortical sensory integration and plasticity (Lee and Sherman, 2010; Smith et al., 2012; Ohga et al., 2018; Lee, 2015; Hu, 2003). While the co-recruitment of lemniscal and non-lemniscal inputs may enhance the generality of our CCK-dependent mechanism, the differing response characteristics of these pathways suggest subtle differences in their relative engagement in the observed plasticity. Future pathway-specific manipulations will help clarify their respective contributions”

Lee, C.C., and Sherman, S.M. (2010). Topography and physiology of ascending streams in the auditory tectothalamic pathway. Proceedings of the National Academy of Sciences 107, 372-377. doi:10.1073/pnas.0907873107.

Smith, P.H., Uhlrich, D.J., Manning, K.A., and Banks, M.I. (2012). Thalamocortical projections to rat auditory cortex from the ventral and dorsal divisions of the medial geniculate nucleus. Journal of Comparative Neurology 520, 34-51.

Ohga, S., Tsukano, H., Horie, M., Terashima, H., Nishio, N., Kubota, Y., Takahashi, K., Hishida, R., Takebayashi, H., and Shibuki, K. (2018). Direct Relay Pathways from Lemniscal Auditory Thalamus to Secondary Auditory Field in Mice. Cerebral Cortex 28, 4424-4439. 10.1093/cercor/bhy234.

Lee, C.C. (2015). Exploring functions for the non-lemniscal auditory thalamus. Frontiers in Neural Circuits 9, 69.

Hu, B. (2003). Functional organization of lemniscal and nonlemniscal auditory thalamus. Experimental Brain Research 153, 543-549. 10.1007/s00221-003-1611-5.

Figure legend section:

“Post-hoc histology at higher magnification (lower-middle) shows the electrode tip confined within the MGv. White lines delineate the MGv/MGd border based on cytoarchitectonic landmarks.”

Statistical Rigor and Data Variability:The remarkably low standard deviations in LTP measurements are unexpected based on established variability in thalamocortical plasticity. The authors' response confirms these values are accurate, but further justification, such as methodological controls or replication-would bolster confidence in these results. Additionally, the comparison of in vivo vs. in vitro LTP variability requires more substantive support.

We appreciate the reviewer's concern regarding the unusually small variability. We would like to clarify that the error bars in our figures represent Standard Error of the Mean (SEM) rather than Standard Deviations (SD). As SEM is derived from the SD while incorporating sample size, it is inherently smaller than SD, which may have led to the impression of unrealistically low variability. This has now been explicitly clarified in the figure legends and Methods.

To illustrate the raw variability, we have added Supplementary Figure 2—figure supplement 1E showing unaveraged fEPSP slopes compare to SEM, corresponding to Figure 2—figure supplement 1C. This addition ensures transparency and allows readers to directly assess the quality and consistency of our recordings.

Regarding the comparison between in vivo and in vitro LTP variability:

We agree that clarifying the basis of our in vivo vs. in vitro variability comparison is important. For example, in Chen et al., 2019, using identical LTP induction protocols (Fig. J), the SED of in vitro slice measurements (Fig. K) was substantially larger than that of in vivo recordings (Fig. L).

This difference likely reflects:

(1) In vitro: neighboring data points within a single experiment are highly correlated; variability across experiments is large due to heterogeneous sensitivity to LTP induction (10–200% increasement).

(2) In vivo: lower correlation between neighboring data points, but each is averaged from 12 recordings over 2 min, reducing cross-trial variability; sensitivity to LTP induction is less variable across experiments (5–60% changes).

We hope that these clarifications and additional data address the reviewer’s concerns regarding statistical rigor and data variability.

Methods section:

“The slopes of the evoked fEPSPs were calculated and normalized using a customized MATLAB script, and the group data were plotted as mean ± Standard Error of the Mean (SEM).”

“All data are presented as mean ± SEM. Error bars and shaded areas represent SEM. Here, n represents the number of stimulation-recording sites or and N represents the number of animals in each experiment. At each time point, fEPSPs were averaged across 12 consecutive trials (2 min) to reduce within-experiment fluctuation. Normalized time courses were then used for repeated-measures analyses.”

Figure legend section:

“Data are mean ± SEM; error bars indicate SEM.”

“(E) Unaveraged fEPSP slopes are shown for each time point, with individual data points corresponding to all sites included in Figure 2—figure supplement 1C; mean ± SEM overlays are shown in black. Note that all individual data points are displayed in this figure, whereas in Figure 2—figure supplement 1C, only the averaged values are shown.”

Viral Targeting and Specificity:The manuscript does not clearly address whether cortical neurons were inadvertently infected by AAV9. Given the potential for off-target effects, explicit confirmation (e.g., microphotograph of stimulation site) would strengthen the study's conclusions.

We appreciate the request for quantitative confirmation of off-target cortical infection. We clarify that our histological verification was conducted by systematic sampling rather than exhaustive quantification. Under the same sampling procedure, we did not detect tdTomato-positive cortical somata after AAV9‑Syn‑ChrimsonR‑tdTomato injections into the MGB, whereas we observed rare EYFP-positive cortical somata after AAV9‑EF1a‑DIO‑ChETA‑EYFP (median < 1 cell per 0.4 × 0.4 mm² section, Figure 2—figure supplement 1F). Although these observations do not constitute a formal statistical estimate, they were consistent across sampled sections and are in line with the low-level trans-synaptic transfer reported for AAV9. We have discussed their potential implications for data interpretation in the Discussion.

We hope these clarifications and the newly presented histological evidence address the reviewer’s concerns and further strengthen the rigor of our study.

Discussion section:

“Another potential limitation of our study is the trans-synaptic transfer property of AAV9 (Figure 2—figure supplement 1F). To mitigate this risk, we carefully control the injection volume, rate, and viral expression time, while also verifying expression post-hoc. Systematic sampling histological analysis detected no tdTomato-positive cortical somata in the ACx (Figure 2E lower panel), whereas rare EYFP-positive cortical somata were observed after AAV9-EF1a-DIO-ChETA-EYFP injections (median < 1 cell in 0.4 × 0.4 mm2 section, Figure 2—figure supplement 1F, corresponds to Figure 2A upper-middle panel). These construct‑dependent observations align with occasional low‑level trans‑synaptic transfer reported for AAV9 (Zingg et al., 2017) and indicate that off‑target cortical infection was negligible for ChrimsonR and exceedingly rare for ChETA under our experimental conditions.”

Zingg, B., Chou, X.L., Zhang, Z.G., Mesik, L., Liang, F., Tao, H.W., and Zhang, L.I. (2017). AAV-Mediated Anterograde Transsynaptic Tagging: Mapping Corticocollicular Input-Defined Neural Pathways for Defense Behaviors. Neuron 93, 33-47. 10.1016/j.neuron.2016.11.045.

Figure legend:

“Representative histological images demonstrating low-level transsynaptic spread following AAV9-EF1a-DIO-ChETA-EYFP injection into the MGv. Rare EYFP-positive cortical neurons were observed (median < 1 cell per 0.4 × 0.4 mm² section). Scale bar: 100 µm.”

Integration of Prior Literature:The discussion of existing work is adequate but could be more comprehensive. A deeper engagement with contrasting findings would provide better context for the study's contributions.

We appreciate the reviewer’s suggestion to engage more deeply with contrasting findings. In the revised Introduction and Discussion, we have:

(1) Refocused the historical context toward adult auditory thalamocortical plasticity and explicitly contrasted it with visual and somatosensory cortices, while adult ACx exhibits weaker and more gated NMDAR dependence.

(2) Positioned CCK–CCKBR signaling as a permissive/gating mechanism that can complement or partially compensate for postsynaptic NMDAR signaling, potentially reconciling variability across cortical areas and life stages.

(3) Clarified the potential differential contributions of lemniscal (MGv) and non‑lemniscal (MGd) streams to plasticity expression and variability, acknowledging pathway-specific response properties.

These additions are now integrated in the Introduction (paragraphs 2–3) and Discussion (sections “CCK Dependence of Thalamocortical Neuroplasticity in the ACx” and “Developmental and Age‑Dependent CCK‑Mediated Plasticity”), providing a more comprehensive and balanced context for our findings.

Introduction section:

“However, converging evidence shows that thalamocortical inputs retain a capacity for experience-dependent modification in adulthood. Sensory enrichment or deprivation can gate or reinstate thalamocortical plasticity. In the adult ACx, pairing sounds with neuromodulatory drive can reshape cortical representations. In vivo high-frequency stimulation (HFS) of dorsal lateral geniculate nucleus (LGN) or medial geniculate body (MGB) induces LTP in sensory cortices and has been linked to perceptual learning beyond the critical period. Notably, auditory thalamocortical plasticity appears less dependent on NMDA receptors compared to other cortical regions. The mechanisms underlying thalamocortical plasticity in the mature brain remain poorly understood.

Cholecystokinin (CCK) and its receptor CCK-B receptor (CCKBR) are well positioned to influence thalamocortical transmission: Cck mRNA is abundant in MGB neurons and CCKBR is enriched in layer IV of ACx, the principal thalamorecipient layer.”

Discussion section:

“These findings suggest a potential involvement of CCK in thalamocortical plasticity. Our data extend this framework by identifying CCK–CCKBR signaling as a permissive modulator of adult thalamocortical LTP.”

“We propose that CCKBR activation may trigger intracellular calcium release and AMPAR recruitment in parallel to, or partially compensating for,independently of postsynaptic NMDAR signaling, while the complementarity of CCKBR and NMDARs may contribute to robust thalamocortical plasticity. This complementary arrangement may reconcile differences across developmental stages and cortical areas, and highlights neuropeptidergic signaling as a lever to re-enable adult thalamocortical plasticity.

Notably, exogenous CCK alone failed to induce LTP in the absence of accompanying stimulation (Figure 5—figure supplement 1A and 2B), emphasizing that CCK function as a modulator rather than a direct initiator of LTP. Activation of the thalamocortical pathway is also essential for LTP induction. Although our experiment targeting the MGv was guided by stereotaxic coordinates and verified post hoc, we acknowledge potential contributions from non-lemniscal medial geniculate nucleus dorsal (MGd) projections. Anatomical and physiological evidence indicates that MGv-AC projections provide rapid, frequency‑specific, tonotopically organized excitation, whereas MGd pathways target higher‑order auditory cortex with broader tuning, less precise tonotopy, longer response latencies, and greater context‑dependence, features that can differentially shape cortical sensory integration and plasticity. While the co-recruitment of lemniscal and non-lemniscal inputs may enhance the generality of our CCK-dependent mechanism, the differing response characteristics of these pathways suggest subtle differences in their relative engagement in the observed plasticity. Future pathway-specific manipulations will help clarify their respective contributions. Another potential limitation of our study is the trans-synaptic transfer property of AAV9 (Figure 2—figure supplement 1F). To mitigate this, we carefully controlled the injection volume, rate, and viral expression time, and conducted post-hoc histological analyses to minimize off-target effects, thereby reducing the likelihood of trans-synaptic transfer confounding the interpretation of our findings.”

Therapeutic Implications:

The authors' discussion of therapeutic potential is now appropriately cautious and well-reasoned.

Conclusion:

While the study presents intriguing findings, the concerns outlined above must be addressed to fully establish the validity and impact of the results. I appreciate the authors' efforts thus far and hope they can provide additional data or clarification to resolve these issues. With these revisions, the manuscript could make a valuable contribution to the field.

**Reviewer #2 (Public review):**
Summary:This work used multiple approaches to show that CCK is critical for long-term potentiation (LTP) in the auditory thalamocortical pathway. They also showed that the CCK mediation of LTP is age-dependent and supports frequency discrimination. This work is important because is opens up a new avenue of investigation of the roles of neuropeptides in sensory plasticity.Strengths:The main strength is the multiple approaches used to comprehensively examine the role of CCK in auditory thalamocortical LTP. Thus, the authors do provide a compelling set of data that CCK mediates thalamocortical LTP in an age-dependent manner.Weaknesses:There are some details that should be addressed, primarily regarding potential baseline differences in comparison groups. The behavioral assessment is relatively limited, but may be fleshed out in future work.

We appreciate the reviewer’s suggestion regarding potential baseline differences. In our study, all groups underwent harmonized procedures, including identical exposure, timing, and acquisition parameters. Group allocation and data collection were performed under standardized conditions. For electrophysiology, baseline fEPSP measures and stimulation intensities were calibrated per site using consistent input-output procedures, with analyses based on normalized slopes relative to each site’s own baseline. For behavior, animals from the same litter served as both experimental and control groups, matched for handling conditions; startle/PPI data were acquired using identical hardware and timing settings. While no additional post hoc re-processing was performed, we have clarified these controls in the Methods to enhance transparency.

We agree that the behavioral assessment is intentionally focused and does not encompass broader auditory perceptual functions (e.g., temporal processing). We now explicitly state this limitation and propose future studies to examine temporal acuity and cell-type-specific manipulations. These experiments will clarify how CCK-dependent thalamocortical plasticity generalizes to other perceptual domains.

**Reviewer #3 (Public review):**
Summary:Cholecystokinin (CCK) is highly expressed in auditory thalamocortical (MGB) neurons and CCK has been found to shape cortical plasticity dynamics. In order to understand how CCK shapes synaptic plasticity in the auditory thalamocortical pathway, they assessed the role of CCK signaling across multiple mechanisms of LTP induction with the auditory thalamocortical (MGB - layer IV Auditory Cortex) circuit in mice. In these physiology experiments that leverage multiple mechanisms of LTP induction and a rigorous manipulation of CCK and CCK-dependent signaling, they establish an essential role of auditory thalamocortical LTP on the co-release of CCK from auditory thalamic neurons. By carefully assessing the development of this plasticity over time and CCK expression, they go on to identify a window of time that CCK is produced throughout early and middle adulthood in auditory thalamocortical neurons to establish a window for plasticity from 3 weeks to 1.5 years in mice, with limited LTP occurring outside of this window. The authors go on to show that CCK signaling and its effect on LTP in the auditory cortex is also capable of modifying frequency discrimination accuracy in an auditory PPI task. In evaluating the impact of CCK on modulating PPI task performance, it also seems that in mice <1.5 years old CCK-dependent effects on cortical plasticity is almost saturated. While exogenous CCK can modestly improve discrimination of only very similar tones, exogenous focal delivery of CCK in older mice can significantly improve learning in a PPI task to bring their discrimination ability in line with those from young adult mice.Strengths:(1) The clarity of the results, along with the rigor multi-angled approach, provide significant support for the claim that CCK is essential for auditory thalamocortical synaptic LTP. This approach uses a combination of electrical, acoustic, and optogenetic pathway stimulation alongside conditional expression approaches, germline knockout, viral RNA downregulation and pharmacological blockade. Through the combination of these experimental configures the authors demonstrate that high-frequency stimulation-induced LTP is reliant on co-release of CCK from glutamatergic MGB terminals projecting to the auditory cortex.(2) The careful analysis of the CCK, CCKB receptor, and LTP expression is also a strength that puts the finding into the context of mechanistic causes and potential therapies for age-dependent sensory/auditory processing changes. Similarly, not only do these data identify a fundamental biological mechanism, but they also provide support for the idea that exogenous asynchronous stimulation of the CCKBR is capable of restoring an age-dependent loss in plasticity.(3) Although experiments to simultaneously relate LTP and behavioral change or identify a causal relationship between LTP and frequency discrimination are not made, there is still convincing evidence that CCK signaling in the auditory cortex (known to determine synaptic LTP) is important for auditory processing/frequency discrimination. These experiments are key for establishing the relevance of this mechanism.Weaknesses:(1) Given the magnitude of the evoked responses, one expects that pyramidal neurons in layer IV are primarily those that undergo CCK-dependent plasticity, but the degree to which PV-interneurons and pyramidal neurons participate in this process differently is unclear.

We agree with the reviewer that the relative contributions of pyramidal neurons and PV-interneurons to CCK-dependent thalamocortical plasticity remain to be determined. Our recordings primarily reflected excitatory postsynaptic activity from layer IV pyramidal neurons, given the fEPSP metrics used. As PV-interneurons are essential in shaping cortical inhibition and temporal precision, they may also be modulated by CCK release from thalamocortical inputs. We have explicitly acknowledged this limitation in the Discussion section of the manuscript and propose that future studies should employ cell-type-specific recording or manipulation approaches to dissect the respective roles of inhibitory and excitatory neuronal populations in CCK-dependent thalamocortical plasticity. We appreciate the reviewer’s suggestion and believe this is a valuable direction for ongoing research.

(2) While these data support an important role for CCK in synaptic LTP in the auditory thalamocortical pathway, perhaps temporal processing of acoustic stimuli is as or more important than frequency discrimination. Given the enhanced responsivity of the system, it is unclear whether this mechanism would improve or reduce the fidelity of temporal processing in this circuit. Understanding this dynamic may also require consideration of cell type as raised in weakness #1.

We acknowledge that the current study primarily examined frequency discrimination and did not directly assess temporal processing. Enhanced network responsivity could have variable effects on temporal precision, depending on the balance between excitation and inhibition. PV-interneurons, in particular, are known to support temporal fidelity in auditory processing (Nocon et al., 2023; Cai et al., 2018). We discussion that future work should investigate how CCK modulation influences temporal coding at both the circuit and single-cell level, and whether such changes align with or diverge from the mechanisms underlying frequency discrimination improvements.

(3) In Figure 1, an example of increased spontaneous and evoked firing activity of single neurons after HFS is provided. Yet it is surprising that the group data are analyzed only for the fEPSP. It seems that single neuron data would also be useful at this point to provide insight into how CCK and HFS affect temporal processing and spontaneous activity/excitability, especially given the example in 1F.

We appreciate the reviewer’s suggestion. While we recorded single-unit activity during HFS protocols, long-term stability over >1.5 hours was less consistent compared to fEPSP measurements, leading to higher variability in spike-based metrics. We therefore used fEPSPs as our primary quantitative measure for robustness. We agree, however, that single-neuron data could yield valuable complementary insights. In future experiments combining stable single-unit recording with synaptic measurements will be conducted to better link cellular excitability and network plasticity.

(4) The circuitry that determines PPI requires multiple brain areas, including the auditory cortex. Given the complicated dynamics of this process, it may be helpful to consider what, if anything, is known specifically about how layer IV synaptic plasticity in the auditory cortex may shape this behavior.

We agree that PPI involves multiple cortical and subcortical nodes. In our paradigm, layer IV neurons receive segregated MGv inputs, high-frequency activation of thalamocortical projections induces robust synaptic plasticity in layer IV. The potentiation at these synapses could amplify the cortical representation of weak prepulses, facilitating their detection and enhancing PPI performance. This interpretation is consistent with prior work showing that local CCK infusion combined with auditory stimuli can augment cortical responses (Li et al., 2014). We have expanded the Discussion to highlight that in aged animals, where baseline PPI performance is often reduced due to degraded auditory inputs (Ouagazzal et al., 2006; Young et al., 2010), restoring thalamocortical plasticity via CCK may partially compensate for sensory gating deficits. We further note that the exact contribution of layer IV to PPI circuitry warrants future investigation using pathway-specific perturbations.

Comments on revisions:The manuscript is much improved and many of the issues or questions have been addressed. Ideally, evidence for the degree of transsynaptic spread for AAV9-Syn-ChrimsonR-tdTomato would also be provided in some form since in the authors' response in sounds like some was observed, as expected.

We thank the reviewer for this important point and for the opportunity to clarify. As requested, we have carefully examined the possibility of transsynaptic spread in our experiments:

We clarify that our histological verification was conducted by systematic sampling rather than exhaustive quantification. Under the same sampling procedure, we did not detect tdTomato-positive cortical somata after AAV9‑Syn‑ChrimsonR‑tdTomato injections into the MGB, whereas we observed rare EYFP-positive cortical somata after AAV9‑EF1a‑DIO‑ChETA‑EYFP (median < 1 cell per 0.4 × 0.4 mm² section, see Figure 2A and Figure 2—figure supplement 1F), consistent with occasional low-level transsynaptic spread reported in the literature.

We have updated the Discussion sections to clearly report these findings, and to emphasize the potential for vector- and construct-dependent variability in transsynaptic spread. We also explicitly acknowledge this technical limitation and discuss its implications for data interpretation.

We hope these clarifications and additions address the reviewer’s concern regarding viral specificity and transsynaptic spread.

Discussion section:

“Another potential limitation of our study is the trans-synaptic transfer property of AAV9 (Figure 2—figure supplement 1F). To mitigate this risk, we carefully control the injection volume, rate, and viral expression time, while also verifying expression post-hoc. Systematic sampling histological analysis detected no tdTomato-positive cortical somata in the ACx (Figure 2E lower panel), whereas rare EYFP-positive cortical somata were observed after AAV9-EF1a-DIO-ChETA-EYFP injections (median < 1 cell in 0.4 × 0.4 mm2 section, Figure 2—figure supplement 1F, corresponds to Figure 2A upper-middle panel). These construct‑dependent observations align with occasional low‑level trans‑synaptic transfer reported for AAV9 (Zingg et al., 2017) and indicate that off‑target cortical infection was negligible for ChrimsonR and exceedingly rare for ChETA under our experimental conditions.”

Zingg, B., Chou, X.L., Zhang, Z.G., Mesik, L., Liang, F., Tao, H.W., and Zhang, L.I. (2017). AAV-Mediated Anterograde Transsynaptic Tagging: Mapping Corticocollicular Input-Defined Neural Pathways for Defense Behaviors. Neuron 93, 33-47. 10.1016/j.neuron.2016.11.045.

Figure legend:

" Representative histological images demonstrating low-level transsynaptic spread following AAV9-EF1a-DIO-ChETA-EYFP injection into the MGv. Rare EYFP-positive cortical neurons were observed (median < 1 cell per 0.4 × 0.4 mm² section). Scale bar: 100 µm."

**Reviewer #1 (Recommendations for the authors):**
Thank you for your efforts in revising the manuscript. While progress has been made, I have a few remaining concerns that I hope you can address to further strengthen the study.Focus of the Introduction:Auditory thalamocortical plasticity is known to be NMDA-dependent, albeit with weaker dependence during early development. Given that this work examines thalamocortical LTP in young adult and aged mice, I recommend refining the Introduction to place greater emphasis on auditory thalamocortical plasticity in the adult brain. The current discussion of somatosensory plasticity during early development, while interesting, seems less directly relevant to the present study. A sharper focus on the auditory system would better frame your research questions.

We thank the reviewer for this constructive suggestion. We have revised the Introduction to emphasize adult auditory thalamocortical plasticity and to streamline content less directly related to our study. Specifically:

(1) We now foreground evidence that thalamocortical inputs retain experience-dependent plasticity beyond the critical period in adult ACx, including neuromodulatory pairing, HFS-induced LTP, and experience-dependent reinstatement.

(2) We explicitly note that adult auditory thalamocortical plasticity is more weakly NMDAR-dependent than in other cortices, thereby motivating our focus on CCK–CCKBR signaling as a permissive mechanism for adult LTP.

(3) We have condensed the discussion of somatosensory plasticity during early development to a brief background and shifted the focus to adult auditory mechanisms and knowledge gaps that directly frame our research questions.

These changes appear in the revised Introduction (paragraphs 2–3), which now provide a sharper rationale for investigating CCK‑dependent thalamocortical LTP in young adult and aged mice.

Introduction section:

“However, converging evidence shows that thalamocortical inputs retain a capacity for experience-dependent modification in adulthood. Sensory enrichment or deprivation can gate or reinstate thalamocortical plasticity. In the adult ACx, pairing sounds with neuromodulatory drive can reshape cortical representations. In vivo high-frequency stimulation (HFS) of dorsal lateral geniculate nucleus (LGN) or medial geniculate body (MGB) induces LTP in sensory cortices and has been linked to perceptual learning beyond the critical period. Notably, auditory thalamocortical plasticity appears less dependent on NMDA receptors compared to other cortical regions. The mechanisms underlying thalamocortical plasticity in the mature brain remain poorly understood.

Cholecystokinin (CCK) and its receptor CCK-B receptor (CCKBR) are well positioned to influence thalamocortical transmission: Cck mRNA is abundant in MGB neurons and CCKBR is enriched in layer IV of ACx, the principal thalamorecipient layer.”

Anatomical Specificity of MGv Targeting:The mouse MGv is a small and deep structure, and precise targeting is critical given the functional differences between MGv and MGd pathways. In the current figures:Fig. 1A suggests the electrode track may have approached the MGd.Fig. 2E indicates some viral spread beyond the MGB.Since MGv-AI and MGd-AC pathways exhibit distinct (and sometimes opposing) effects on plasticity, I encourage you to provide additional clarification or verification of the stimulated/infected regions. This would greatly enhance the interpretability of your LTP data.

Please see above.

Data Variability and Transparency:The reported thalamocortical LTP values exhibit remarkably small standard deviations, which is somewhat unexpected given typical experimental variability in such measurements. To address this concern, it would be helpful to include example raw traces of the recorded LTP (e.g., in a supplementary figure). This would allow readers to better evaluate the data quality and consistency.

Please see above.

**Reviewer #2 (Recommendations for the authors):**
Overall, the authors did an excellent job of responding to our critiques, both in their direct responses and in the modified text. The modified text is also more readable than before. Two issues that the authors should consider addressing;(1) Unless I missed it, there is no commentary stated about the impact of using aged C57 mice, which lose their hearing, such that the effects seen in the older mice could be related to hearing loss rather than aging alone. Some discussion of this point should be made.

We thank the reviewer for raising this important point. C57BL/6 mice are known to develop age-related hearing loss, which could potentially affect PPI performance in older animals. We note that in our internal screening we observed markedly reduced startle amplitudes and frequent negative PPI values in many mice >20 months, indicating severe auditory impairment. To minimize this confound a priori, we excluded mice older than 20 months and restricted the aged cohort to 17–19 months, which consistently exhibited robust startle responses and reliable PPI. While some degree of presbycusis may still be present in this age range in C57BL/6 mice, the improvement of PPI following CCK administration combined with acoustic exposure indicates that the auditory pathways remained sufficiently functional to support sensorimotor gating. In fact, the presence of partial hearing loss in these aged mice may have allowed us to better detect the beneficial effects of CCK, further highlighting its therapeutic potential for age-related deficits. The greater improvement in PPI observed in older mice —as compared to younger mice, whose PPI in control group is already high—likely reflect the combined effects of age-related hearing loss and CCK deficiency, with CCK-induced restoration of thalamocortical plasticity being the primary focus of our study. We have now added a discussion of this point in the revised manuscript.

Discussion section:

“In aged mice, PPI deficits are commonly observed due to impaired auditory processing. Notably, C57BL/6 mice exhibit age-related hearing loss (Johnson et al., 1997). Both age-associated changes in auditory function and CCK deficiency contribute to impaired sensory gating. The presence of partial hearing loss in aged mice may have facilitated the detection of CCK’s beneficial effects, further highlighting its therapeutic potential for age-related deficits. Our results suggest that enhanced thalamocortical plasticity mediated by CCK might partially compensate for these deficits by amplifying residual auditory signals in aged mice.”

Johnson, K.R., Erway, L.C., Cook, S.A., Willott, J.F., and Zheng, Q.Y. (1997). A major gene affecting age-related hearing loss in C57BL/6J mice. Hearing Research 114, 83-92. https://doi.org/10.1016/S0378-5955(97)00155-X.

(2) Minor point - I do not agree with the use of the term "ventral to bregma" to describe where the craniotomies were placed (e.g., line 599). The direction being described is more typically referred to as "lateral." If the authors prefer to use the term "ventral," perhaps additional clarification can be added.

We thank the reviewer for pointing out this issue and apologize for any confusion. We agree that “ventral to bregma” is not the standard terminology and have revised the Methods section to use “below the temporal ridge”. We have also clarified that the craniotomy for accessing the auditory cortex was performed on the lateral aspect of the skull in rodents, just below the temporal ridge. We hope this revision resolves the ambiguity.

Method section:

“A craniotomy was performed over the temporal bone, as the auditory cortex is located on the lateral surface of the brain (coordinates: 1.5 to 3.0 mm below the temporal ridge and 2.0 to 4.0 mm posterior to bregma for mice; 2.5 to 6.5 mm below the temporal ridge and 3.0 to 5.0 mm posterior to bregma for rats) to access the auditory cortex.”

“Six-week after CCK-sensor virus injection, a craniotomy was performed to access the auditory cortex at the temporal bone (1.5 to 3.0 mm below the temporal ridge and 2.0 to 4.0 mm posterior to bregma), and the dura mater was opened.”